# An electroluminescent and tunable cavity-enhanced carbon-nanotube-emitter in the telecom band

Anna P. Ovvyan[1], Min-Ken Li[2,3], Helge Gehring [1], Fabian Beutel[1],
Sandeep Kumar[4], Frank Hennrich[2], Li Wei [5], Yuan Chen [5], Felix Pyatkov[3,4],
Ralph Krupke[2,3,4] & Wolfram H. P. Pernice [1,6,7] ✉

Emerging photonic information processing systems require chip-level integration of controllable nanoscale light sources at telecommunication wavelengths. Currently, substantial challenges remain in the dynamic control of the sources, the low-loss integration into a photonic environment, and in the site-selective placement at desired positions on a chip. Here, we overcome these challenges using heterogeneous integration of electroluminescent (EL), semiconducting carbon nanotubes (sCNTs) into hybrid two dimensional – three dimensional (2D-3D) photonic circuits. We demonstrate enhanced spectral line shaping of the EL sCNT emission. By back-gating the sCNT-nanoemitter we achieve full electrical dynamic control of the EL sCNT emission with high on-off ratio and strong enhancement in the telecommunication band. Using nanographene as a low-loss material to electrically contact sCNT emitters directly within a photonic crystal cavity enables highly efficient EL coupling without compromising the optical quality of the cavity. Our versatile approach paves the way for controllable integrated photonic circuits.

Electrically-driven light sources with a nanoscale footprint are desirable for integrated photonic circuits since they avoid illumination with an excitation laser and the accompanying high-fidelity optical filtration of the pump light on the chip, thus reducing design complexity substantially. Semiconducting carbon nanotubes (sCNT) have emerged as promising truly nanoscale light sources that can be electrically stimulated into stable excitonic emission[1]. sCNTs show quantum behavior at cryogenic[2–4] and at room temperature[5] as a result of localization of excitons at defects. A first limitation for realizing scalable on-chip circuits is the site-selective placement of a nanoscale source at the desired location on a photonic device, which is particularly important when aiming at integrating emitters into photonic cavities. Dielectrophoresis provides such a stable deposition method for sCNTs[6] and

allows deterministic placement between electrodes. Secondly, the integration of electrically-driven sCNT with nanophotonic waveguides requires suitable current paths for electrical stimulation. This is not straightforward, because traditionally-used metallic electrodes induce optical absorption loss at the waveguide, leading to a substantial decrease of the coupling efficiency of emitted light into the propagating mode. Furthermore, metal electrodes suppress and degrade resonance modes of nearby photonic cavities[7], which are convenient for spectral filtering and Purcell enhancement of emitted light. Besides, metal electrodes situated nearby the sCNT leads to broadening of the sCNT emission[8]. We overcome these challenges by using waveguide-compatible, transparent and low-loss nanocrystalline graphene (NCG) electrodes instead of metallic electrodes[9–11]. sCNT

[1]University of Münster, Physikalisches Institut, Center for Nanotechnology, Heisenbergstr. 11, 48149 Münster, Germany. [2]Institute of Quantum Materials and Technologies, Karlsruhe Institute of Technology, 76021 Karlsruhe, Germany. [3]Institute of Materials Science, Technische Universität Darmstadt, 64287 Darmstadt, Germany. [4]Institute of Nanotechnology, Karlsruhe Institute of Technology, 76021 Karlsruhe, Germany. [5]The University of Sydney, School of Chemical and Biomolecular Engineering, Darlington, NSW 2006, Australia. [6]Center for Soft Nanoscience, Busso-Peuss-Str. 11, 48149 Münster, Germany. [7]Kirchhoff-Institut for Physics, Im Neuenheimer Feld 227, 69120 Heidelberg, Germany. ✉e-mail: wolfram.pernice@kip.uni-heidelberg.de

heterojunctions enable the controlled injection of electrons and holes into a CNT and the generation of narrowband electroluminescence (EL)[12]. We embed NCG-electrodes directly within a photonic crystal (PhC) cavity with negligible optical insertion loss and no degradation of the Q-factor of the resonance mode. We employ (9,8)-sCNTs, which emit in the telecommunication E-band with an EL exitonic peak intensity around 1440 nm[8,9]. In order to increase the quantum yield of sCNT[13,14], we engineer a suitable photonic environment with enhanced Local Density of States (DOS) to raise the spontaneous emission rate of the incorporated sCNT (9,8) according to Purcell theory[15].

We demonstrate enhancement of the EL emission of sCNTs by efficient coupling into a hybrid nanographene-PhC cavity device. Importantly, we obtain full dynamic control of the intensity of the enhanced EL sCNT by active electrical operation of the back-gate voltage, which in fact proves the electroluminescent nature of the emitted sCNT light. Our hybrid approach enables active control of the intensity of narrow-linewidth EL spectra via modification of the excitonic (sCNT) emission by electrical operation via in particular utilization 2D nanocrystalline graphene electrodes integrated into a tailored 3D photonic structure. The applied low driving bias-current (tens of nanoamperes) to the sCNT further proves the electroluminescent nature of the emitted light, in contrast to previous work by Pyatkov et al. [7] where three orders of magnitude higher biasing current was utilized to generate incandescent emission of CNTs. We achieve an *on-off* ratio close to 100%. In the *switched-on* regime, we find a high enhancement factor up to $F_{int} = 188$ and coupling efficiency $\beta_{int} = 99.5\%$ of the EL into the fundamental resonance mode, which is efficiently read out by 3D couplers terminating the PhC cavity. This way, we achieve high coupling efficiency of telecom electroluminescence in a tailor-made 1D PhC cavity mode in contrast to collecting photoluminescence from sCNTs in a cavity[16] and exciton-plasmon coupling[17].

In addition, we show that nanocrystalline graphene strip in the cavity region can function as thermal nanoemitter, thus providing telecom-band polarized emission with a peak enhancement factor $F = 80$ and $F = 112$ at 77 K and 300 K, respectively. Such hybrid devices allow to experimentally study the Local Density of States (LDOS) in the cavity region, thus determining optimal spatial point of efficient emitter coupling into resonance modes. The nanocrystalline graphene strip incandescent emitter can be universally utilized for on-chip optical LDOS probing at room and cryogenic temperature.

The electrically controllable hybrid NCG-Si$_3$N$_4$ photonic circuits with deterministic placement of sCNTs in the cavity region provide a scalable and reproducible platform which meets the requirements of integrated photonic circuits for classical and quantum applications.

## Results

### Hybrid cross-bar PhC cavity device
We design cross-bar PhC cavities with embedded NCG nanoelectrodes (Fig. 1) in order to electrically control the enhanced emission rate of cavity-integrated light sources. The cross-bar design allows to position NCG electrodes directly atop Si$_3$N$_4$ waveguides, while the enhanced EL optical signal is outcoupled through another waveguide using 3D couplers[18]. The employed total-internal reflection couplers provide convenient and broadband out-of-plane readout of the optical signal (Fig. 1b) with good efficiency.

The devices were realized on Si$_3$N$_4$/SiO$_2$ on Si substrates using a multi-step electron beam lithography protocol. Following planar fabrication of the nanophotonic circuits, the 3D couplers are printed on the chip, subsequently. A helium ion microscope (HIM) image of one of the couplers is shown in Fig. 1b. In the final fabrication step nanophotonic devices were equipped with poly-(9,9-di-n-dodecylfluorenyl-2,7-diyl)-wrapped (9,8)-sCNT (central wavelength 1440 nm)[8,19] deposited in between nanocrystalline graphene nanoelectrodes via site-selective dielectrophoresis[12]. This fabrication procedure allows realization of scalable and reproducible hybrid devices on chip, as

discussed in the Methods section. The cross-bar PhC cavity enhances the spontaneously emitted EL of sCNT integrated in the cavity region and further enables its direct coupling to nanophotonic waveguides, thus making it compatible with further integrated optical nodes in the circuit as shown in Fig. 1a, d, e. The light-matter interaction strengthens the radiative emission rate of sCNT by Purcell enhancement[15], while at the same time the PhC cavity provides narrow-line spectral filtering of the emitted light. To minimize the influence of the crossed bar on the quality factor (Q factor) of the cavity, the arm is linearly tapered down from a width of 600 nm to 200 nm (Fig. 1d, Fig. 2, $w_{cr}$ in Fig. 3c) to the cavity region along 1 μm. Figure 1b in the supplementary information (SI) shows good agreement of the simulated and measured Q-factors of a range of fabricated cross-bar PhC devices. The electric field distribution of additional odd resonance modes is shown in Supplementary Fig. 4b. We employ 3D Finite Difference Time Domain (FDTD) numerical optimization of the PhC cavity to maximize the coupling efficiency of emitted light into resonance modes (β-factor). The β-factor correlates with the enhancement factor and also the optical transmission through the device. Maximum enhancement is obtained by placing the emitter within the antinode of the electric field according to the Purcell theory[15]. Notably, our hybrid PhC design (Fig. 1a, d, e) naturally allows to obtain high enhancement factors because the polarization of the sCNT EL is aligned with the TE-like resonance mode of the cavity. The increase of the spontaneous emission rate leads to an increase of the β-factor coupling into the corresponding mode, which is the ratio of enhanced emitted light outcoupled from both ends of the cavity to the total amount of light emitted by the source.

The optimal position of the source within the cavity region is determined via 3D FDTD simulations of the LDOS enhancement. The results for different positions of the emitter on top of the cavity region along longitudinal and transverse directions are summarized in on-resonance LDOS spatial maps shown in Fig. 4a and Supplementary Fig. 4a, respectively. The electroluminescent sCNT emitter is modelled as a classical dipole source with an electric field aligned along the TE mode of the cavity. Light emitted from the source, placed on the center of the cavity, is coupled to odd resonance modes and enhanced. Moving the emitter away from the cavity center along longitudinal and transverse directions leads to a decrease of the enhancement of odd modes. Thus, the LDOS-spatial maps qualitatively indicate an optimal central position of sCNTs in the cavity region to obtain the highest enhancement, where the coupling efficiency into odd modes is maximized. This is in agreement with the corresponding electric field distribution of the modes (Supplementary Fig. 4b). Controllable integration of a single sCNT in the required location is achieved by self-alignment between electrodes through the induced electrostatic field during dielectrophoresis. Thus, owing to the design, a single sCNT is placed in the optimal position in the center of the cavity as shown in Fig. 1d (see inset).

### Low-loss cavity-compatible nanographene electrodes
A key ingredient to our approach consists of the engineered nanocrystalline graphene-sCNT heterojunction atop the cavity region of the cross-bar PhC. The electrodes located on the crossed bar are required to reach closely into the center of the cavity to drive the sCNT (Fig. 1a, c, d, e) and thus they potentially can affect the resonance modes. Therefore, we replace traditional absorptive metal electrodes with nanocrystalline graphene nanoelectrodes atop of the cavity.

The benefit of using nanocrystalline graphene electrodes compared to gold electrodes is shown in Fig. 2. The measured insertion loss of the crossed structure increases with increasing width of the cross-bar as shown in Fig. 2a. While insertion losses at junctions without electrodes are as low as 0.046 ± 0.006 dB/cross (red data points), they reach 0.084 ± 0.007 dB/cross for nanocrystalline graphene electrodes (green data points) and 1.050 ± 0.021 dB/cross for gold electrodes

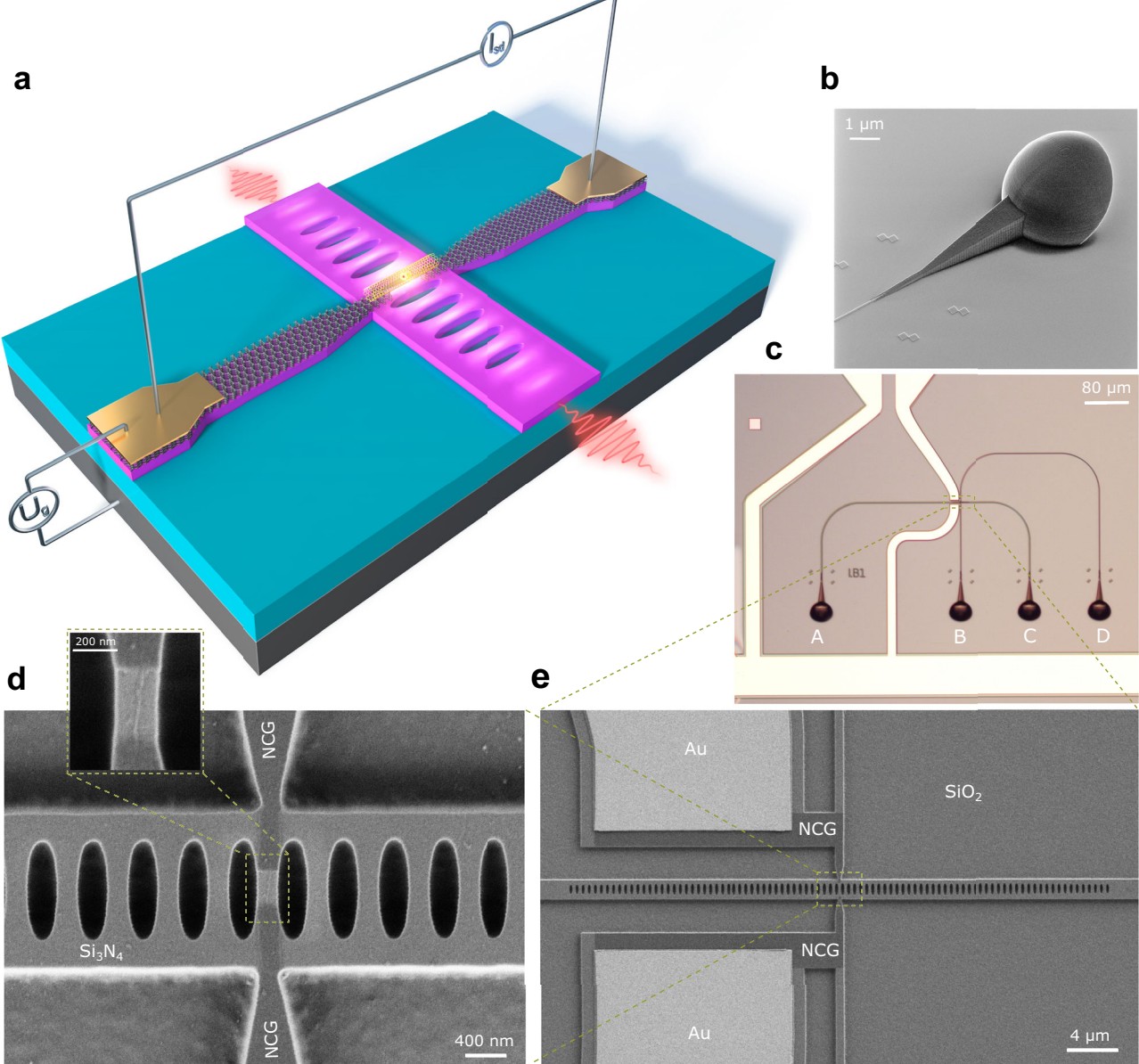

**Fig. 1 | Electrically controlled cross-bar photonic crystal (PhC) cavity with integrated semiconducting carbon nanotube (sCNT) emitter. a** Schematic of the hybrid device. The sCNT is positioned between nanocrystalline graphene electrodes placed on top of the cavity crossing. The nanographene electrodes are used for addressing the sCNT and are connected to metallic leads for current injection. The electroluminescent emission is coupled out from both ends of the PhC cavity. **b** Helium Ion microscopy image of a broadband total reflection three-dimensional

(3D) coupler connected to the ends of the nanophotonic waveguides. **c** Optical micrograph of the fabricated device with 3D output couplers and cavity region marked by the zoom-box. Output ports are marked as A-D. **d** Scanning electron microscope (SEM) image of the PhC cavity with a single sCNT integrated between nanocrystalline graphene electrodes. The inset shows a close-up image of an individual sCNT in the cavity. **e** Scanning electron microscope image of the hybrid device in the cavity region.

(blue data points) for a cross-bar width of 200 nm at a wavelength of 1550 nm. We note that insertion loss at a wavelength of 1490 nm shows a similar trend.

The effect of the electrode material on the Q-factor of a cross-bar PhC cavity is shown in Supplementary Fig. 2. We find that influence of nanocrystalline graphene electrodes on Q-factor is negligible, which can be seen by comparing the green and red curves, corresponding to similar cavities with NCG and without deposited electrodes, respectively. In contrast, gold electrodes decrease the Q-factor of the resonance mode by more than 4.9-7.8 times, depending on the cross width, and also suppress transmission of odd modes, because the antinodes of their electric fields are located in the center of the cavity. Increasing the width of the cross-bar $w_{cr}$ leads to a decrease in Q-factor for all

considered cases. Utilizing graphene electrodes for the excitation of sCNT provides improved electrical contact and electric field distribution for dielectrophoresic deposition. The incorporation nanocrystalline graphene electrodes lowers bending at the contact edge which may avoid unwanted defects[20], while the heterojunction improves the overall transport behaviour.

## NCG-based incandescent nanoemitter in the telecom band

Besides functioning as driving electrodes, we integrate a nanocrystalline graphene strip with a narrow junction between NCG electrodes within the cavity center of PhC to form an incandescent emitter[9], as shown in Fig. 3a, c complementing the electroluminescent sCNT emitter. Such a thermal strip-nanoemitter incorporated in a

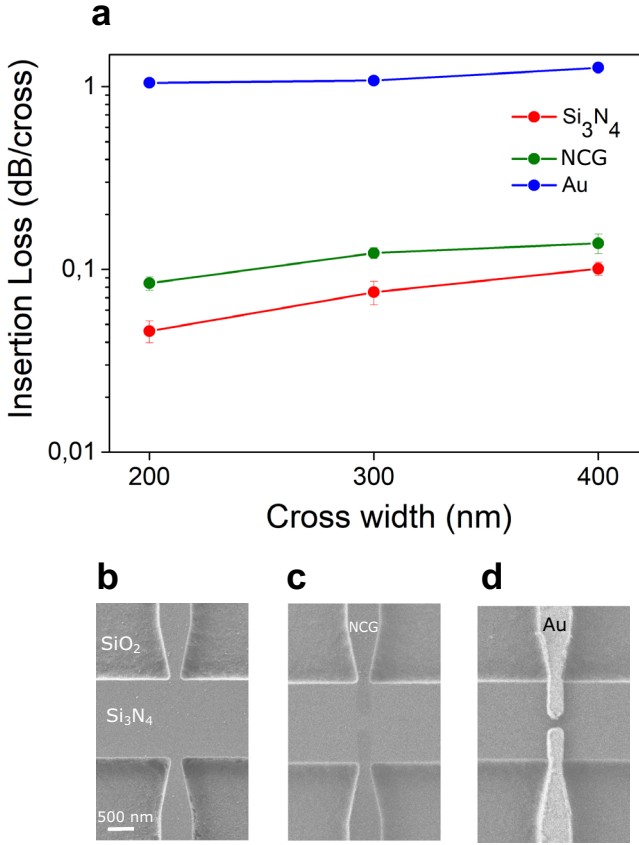

**Fig. 2 | Low-loss nanocrystalline graphene electrodes embedded in a photonic crystal (PhC) cavity. a** Measured insertion loss of cavity cross-bar structures with integrated nanocrystalline graphene (NCG) electrodes (green curve), gold electrodes (blue curve), without electrodes (red curve). The error bar was calculated by measuring the total loss through a device with multiple electrodes in dependence of the number of electrodes and fitting the result by linear regression. Scanning electron microscope (SEM) images of cavity cross-bars with a tapered width 200 nm (**b**) without electrodes, (**c**) with 5 nm thick NCG electrodes and (**d**) 120 nm thick gold electrodes. The scale bar represented in panel (**b**) applies also to panels (**c, d**).

predetermined position within a photonic device allows to probe the LDOS factor and provides optimal coupling of emitted light into cavity modes at cryogenic and room temperature. The nanocrystalline graphene strip nanoemitter can be removed by oxygen plasma (while protecting the rest of the circuit with photoresist, for example) and thus makes it a promising candidate for optical high-resolution LDOS sensing at room and cryogenic temperature. Importantly, the NCG strip-nanoemitter does not degrade the quality factor of the cavity as shown in Supplementary Fig. 2c in contrast with the work of Shiue et al. [21], where a micro black-body radiator for optical communication applications was demonstrated.

The spectral properties of nanostrip devices are determined in a custom imaging system as described in the Methods section. We measure the outcoupled enhanced resonance mode spectrum from electrically stimulated biased nanocrystalline graphene strip at 77 K at the read-out couplers A and C with an InGaAs photodiodes array as shown in Fig. 3d. We emphasize that in the case of the demonstrated biased NCG nano-strip emitter, the supplied electric energy is transformed into Joule heat and is dissipated in the nanocrystalline graphene leading to incandescent emission. The estimated electron temperature of the NCG-based strip-emitter on the $Si_3N_4$ waveguide at an applied electrical power of 2.87 mW (source-drain current 70 uA) results in $T_e$ of ~1000 K, as shown in Supplementary Fig. 8 and

discussed in the Supplementary Section 8. Thus, the electrical excitation of the thermal strip-nanoemitter (hundreds of microamperes) is three-four orders of magnitude higher in comparison to the electroluminescent sCNT reported in the next paragraph, due to the incandescent nature of the NCG emission. Thermal radiation of the electrically-biased NCG nano-strip source is confirmed by the measured independence of the source-drain current through the NCG during the change of the back-gate voltage, shown in Supplementary Fig. 9.

The central position of NCG strip (Fig. 3a) provides optimal coupling of emitted incandescent light into odd resonance modes (narrow linewidth peaks in red and green spectra in Fig. 3d), since the antinodes of their electric field distributions are concentrated in the center, as predetermined by computed LDOS-spatial maps (Fig. 4a and Supplementary Fig. 4a). The measured spectrum depicted in Fig. 3d is in good agreement with the simulated spectrum shown in Fig. 3b. A free-space confocal emission spectrum of light detected from the NCG strip is shown in blue. The dips at the resonance wavelength confirm the effective coupling of incandescent light into the resonance modes of the PhC cavity. The envelope of the spectrum is modulated by substrate-induced interference[22].

A polarization map at 77 K as depicted in Fig. 3e proves that the incandescent light obtained from couplers A and C is strongly polarized, where the emission shows a maximum at a polarization angle of 90° and vanishes almost completely at 180°. High polarization contrast is an indicator of successful coupling of NCG strip thermal emission into the PhC cavity, which can be further characterized by suitable figures of merit, such as the enhancement factor and the coupling efficiency.

The cross-bar PhC cavity enables strong modification of NCG incandescent emission at cryogenic and at room temperature (as discussed in Supplementary Information and shown in Supplementary Fig. 3b). Therefore, further measurements are performed in both regimes, at 77 K and 300 K, respectively. The cross-bar PhC cavity provides on-resonance enhancement of NCG incandescent emission evanescently coupled into odd modes, obtaining maximum values $F_{III} = 80.6$ at cryogenic and $F_{III} = 112.7$ and room temperature. These factors are experimentally determined as the ratio of the on-resonance intensity of light outcoupled from couplers A and C to the free-space intensity extracted from the NCG strip at the corresponding resonance wavelength. Furthermore, we determine the on-resonance coupling efficiency (β-factor) as the ratio of enhanced emitted light at resonance wavelength detected from couplers A and C and all thermal light emitted by graphene strip, namely the sum of intensity of light detected confocally at thermal source and at the couplers, resulting in a peak value $β_{III} = 98.7\%$ at 77 K and $β_{III} = 99.1\%$ at 300 K.

## Electrically controlled electroluminescence from a sCNT

In order to electrically control the emitter, we embed single (9,8)-CNTs in between optimized nanocrystalline graphene electrodes coupled to a cross-bar PhC cavity. The cavity region of the experimentally investigated device is shown in Fig. 1d. The NCG electrodes serve as source and drain electrodes and the silicon substrate underneath the dielectric as the back-gate. The applied back-gate voltage produces a vertical electric field between the sCNT and the Si substrate to control the charge in the sCNT channel. In this field-effect configuration photons will be emitted from an electrically driven sCNT, provided that electrons are injected at one CNT/NCG contact and holes at the other contact in such a way that they form an exciton in the nanotube channel and radiatively recombine. Such an excitonic EL emission is obtained when the sCNT channel is charge-neutral and the influx of electrons and holes into the channel is comparable. For our device, this is the case when the source-drain voltage is maximal (see Fig. 5b). This scenario corresponds to the EL *switched-on* regime. The charge in the sCNT is not neutral anymore, when the source-drain voltage is

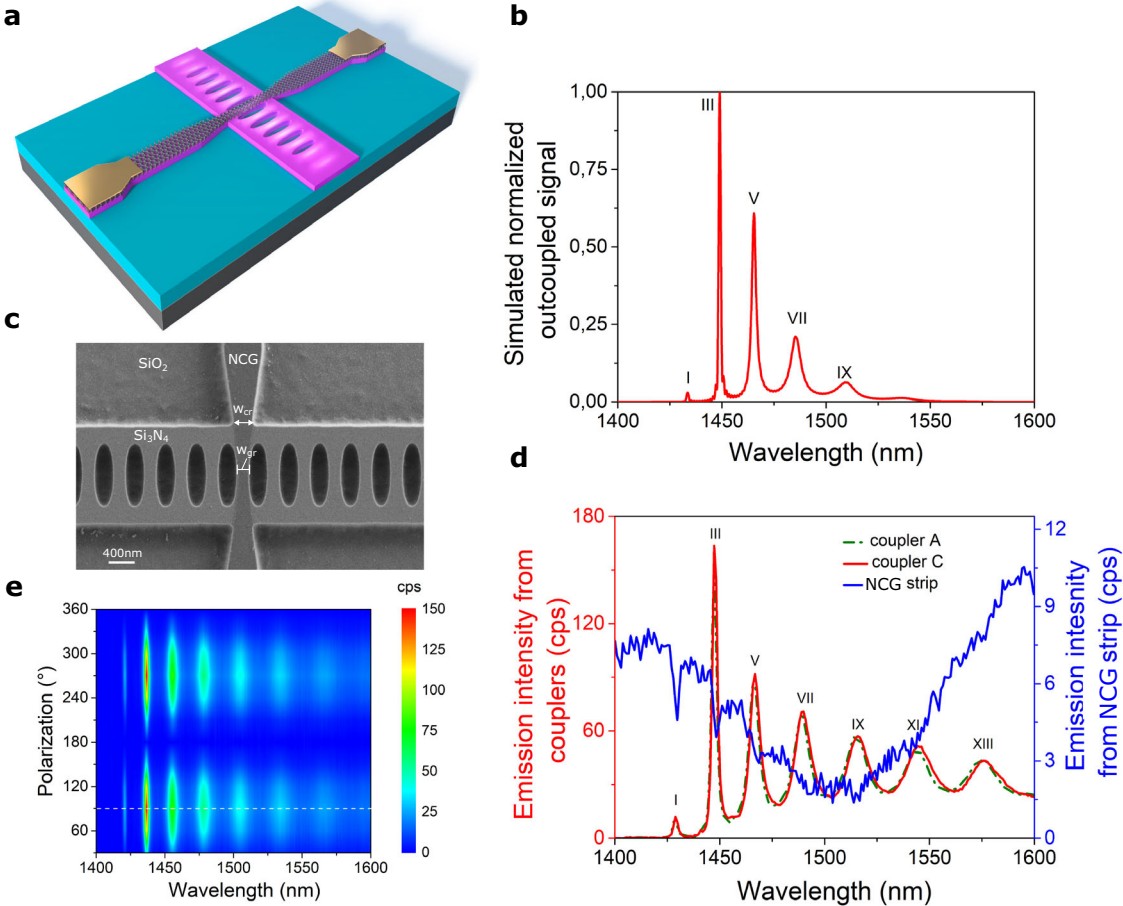

**Fig. 3 | Nanocrystalline graphene-based emitter. a** Schematic of the hybrid device. Nanocrystalline graphene strip integrated atop cross-bar PhC cavity. **b** Three-dimensional finite difference time domain (3D FDTD) simulated spectrum (normalized) of light outcoupled from one of the ends of the cross-bar photonic crystal (PhC) cavity. The symmetric design of the PhC cavity leads to equal incandescent emission at both sides. The resonance modes are labeled in roman numbers (I–IX) in the plot. **c** Scanning electron microscope (SEM) image of the cavity region with NCG strip atop cavity. The waist of NCG strip is 250 nm in width, thickness – 5 nm. **d** Recorded spectra of enhanced thermal emission outcoupled from couplers A (green curve) and C (red curve) of an electrically-biased NCG strip

using 120uA (source-drain voltage 35.5 V). The spectrum from the nanocrystalline graphene (NCG) strip (blue curve) was acquired with the polarizer parallel to TE mode of the waveguide. The enhancement factors and coupling efficiency of incandescent emission into the I, III, and V resonance modes are $F_I = 5.3$, $F_{III} = 80.6$ and $F_V = 58.7$, $\beta_I = 84.1\%$, $\beta_{III} = 98.7\%$, and $\beta_V = 98.3\%$, respectively. **e** Measured spectra of the incandescent emission detected at coupler C, projected onto different polarization angles with a polarization filter. The dashed white line marks the polarization axis of the spectra acquired at couplers A and C. The measurements were performed at 77 K. The simulated and fabricated cross-bar PhC cavity contains $N = 45$ segments in each Bragg mirror with a lattice period of $a = 458$ nm.

minimal. In this case the excitonic emission is suppressed, corresponding to the EL *switched-off* regime. We demonstrate that the EL emission can be switched by a gate-voltage and by using current-biasing instead of voltage-biasing, the gate-voltage range for the excitonic EL emission becomes independent from the bias value, and that the emission then becomes highly stable[8]. Thus, in order to control the quantum yield of excitonic emission[8,9,12], gating of the tube in a back-field configuration is performed producing an electric field between the silicon substrate and the sCNT (without dissipation of energy) as shown in Fig. 1a. This allows us to dynamically control the intensity of the fundamental resonance mode which is driven by electroluminescent, which testifies the EL nature of the emission (in contrast with incandescent emission). The collected light in the waveguides is again outcoupled by 3D couplers (Fig. 1a–c) and detected by an InGaAs photodiode linear array.

Biasing the sCNT integrated in the cross-bar PhC cavity with simultaneously applied back- gate voltage ($U_g = 20$ V) leads to an enhanced excitonic emission coupled into the fundamental resonance mode, where the simulated spectrum of light coupled out from one of the ends of the investigated PhC is shown in Fig. 5a. The measured spectrum outcoupled from coupler C is shown by the red curve in

Fig. 5d. This regime corresponds to the EL *switched-on* state. Importantly, the utilized biasing current of the sCNT is three-four orders of magnitude lower in comparison with the nanocrystalline graphene strip thermal emitter described in the previous paragraph, as well as in comparison with the incandescent CNT realized in the work of Pyatkov et al. [7]. This again confirms the EL nature of our sCNT emission in contrast with incandescent NCG emission.

The presence of the EL signal at the end of the PhC cavity (red curve in Fig. 5d) and the simultaneous absence of the free-space EL signal detected at the position of the investigated coupled sCNT (green curve in Fig. 5d) in the *switched-on* state ($U_g = 20$ V) of the demonstrated device confirm successful coupling of EL into the fundamental resonance mode of the PhC cavity.

The central orientation of NCG nanoelectrodes and sCNT ensures efficient coupling of the emitted light preferentially into the odd resonance modes due to a locally enhanced density of states (Fig. 4a and Supplementary Fig. 4a). Turning off the gate voltage results in strong suppression of the excitonic emission, the EL *switched-off* state of the mode (blue curve) in Fig. 5d.

In order to demonstrate full control of the excitonic emission we acquire an electroluminescent excitation map shown in Fig. 5e where

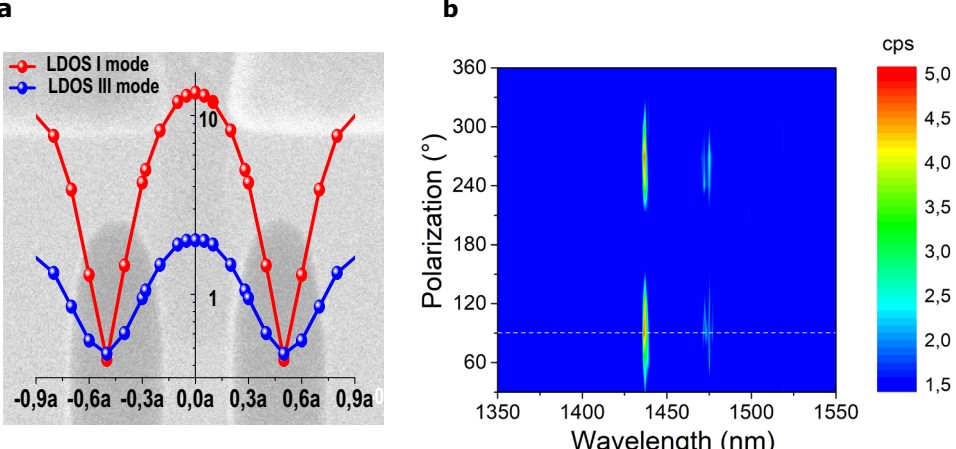

**Fig. 4 | Local density of states (LDOS) enhancement spatial map and polarization map. a** The simulated LDOS enhancement spatial map of emitted electroluminescence (EL) on resonance from a semiconducting carbon nanotube (sCNT) placed between nanocrystalaline graphene (NCG) electrodes in a cross-bar photonic crystal (PhC) cavity. The PhC contains $N = 25$ holes in each Bragg mirror with a lattice period of $a = 462$ nm. The emitter is positioned in the cavity center and varied along the longitudinal direction. The position of the source along the x-direction is plotted normalized to the period a. **b** Spectra of EL emission detected at a three-dimensional (3D) coupler C, projected onto different polarization angles. The white dashed line marks the polarization axis of all experimentally acquired spectra.

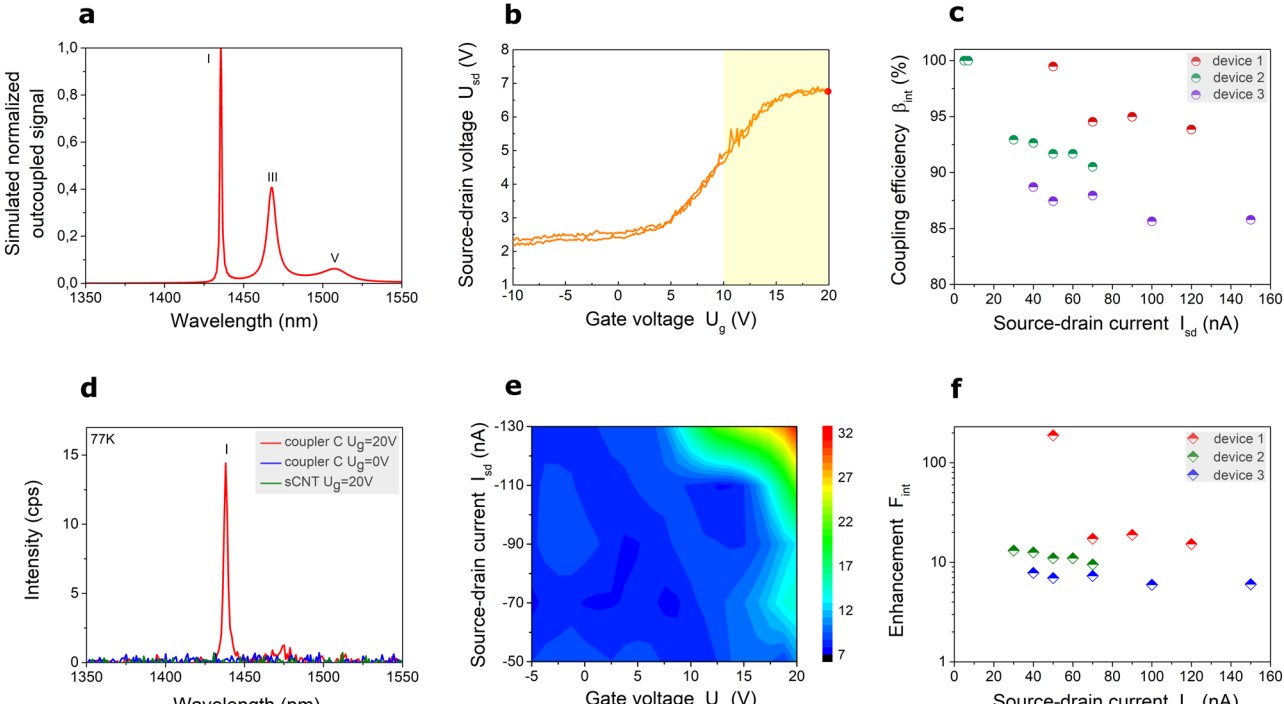

**Fig. 5 | Dynamic control of electroluminescence (EL) from cavity-integrated semiconducting carbon nanotube (sCNT). a** Simulated spectrum of light coupled out from one of the ends the cross-bar photonic crystal (PhC) cavity. The symmetric design of the PhC cavity leads to equal EL emission at both sides. **b** $U_{sd}$-$U_g$ curve acquired at constant sCNT biasing current $I_{sd} = -90$ nA. Forward and backward sweep traces are shown. The yellow area corresponds to the regime, in which the excitonic EL emission is in the *switched-on* state. The red point corresponds to the electric acquisition point of the spectrum (red curve) shown in (**d**). **c** Measured dependence of full-bandwidth coupling efficiency $\beta_{int}$ on $I_{sd}$ biased sCNT for several PhC devices. **d** The spectrum of EL sCNT coupled to the fundamental resonance mode at 1438.1 nm, acquired from coupler C of a PhC cross-bar device gated with gate-voltage $U_g = 20$ V (EL *switched-on* state, red curve*)* and at $U_g = 0$ V (EL *switched-off* state, blue curve). The spectrum of free-space EL simultaneously recorded at the position of the sCNT coupled to the cavity in the *switched-on* state at $U_g = 20$ V of the same investigated device is shown by the green curve. sCNT biasing current is $I_{sd} = -90$ nA. **e** EL excitation map of enhanced EL intensity integrated over the fundamental resonance mode (1430–1450 nm) outcoupled from coupler C of explored device as a function of driven source-drain current $I_{sd}$ and gate voltage $U_g$. **f** Measured dependence of full-bandwidth enhancement-factor $F_{int}$ on $I_{sd}$ applied to the sCNT for three cross-bar PhC devices, acquired at 77 K. Device 1 consists of $N = 25$ segments in each Bragg mirror with a lattice period of $a = 462$ nm; device 2: $N = 35$, $a = 460$ nm; device 3: $N = 35$, $a = 455$ nm. All data recorded at 77 K.

the integrated EL intensity of the sCNT coupled into fundamental resonance mode and detected at coupler C - is plotted as a function of source-drain current-bias and back-gate voltage. Increasing the biased source-drain current leads to light emission at large positive gate voltage, which can be understood from the charge-transport measurement of the sCNT. Figure 5b shows the source-drain voltage that is required to maintain the fixed source-drain current as a function of gate voltage. At large positive gate voltage, the required voltage is large and this corresponds to the regime when the electron and hole currents are low but similar. As described above this is the condition for exitonic EL emission and yields the EL *switched-on* state. The transport data shows negligible hysteresis between the forward and backward sweeps due to the cryogenic environment (77 K)[8]. In addition, we demonstrate the EL *switched-on* state also at room temperature as depicted in Supplementary Fig. 5b, resulting in enhanced narrow linewidth EL coupled into fundamental resonance peak in the spectrum. Notably, the sCNT (9,8) EL linewidth at room temperature is 50 nm (30 meV), which reduces to 20 nm (12 meV) at 77 K[8]. The EL linewidth is considerably wider than the linewidth of the PhC cavity (1.7 nm), which illustrates successful coupling of EL into the mode and suppression of emission outside the cavity resonances.

As a proof of principle of our versatile approach of full electrical control of an EL-sCNT, we demonstrate in the Supplementary Section 6 the experimental measurement of another hybrid device where the enhanced EL-sCNT (which is coupled to a low-loss nanographene-photonic environment) is dynamically operated via back-gate voltage regulation.

The polarization-dependent measurement of the cavity emission is shown in the acquired map as a function of the polarization angle of the inserted polarizer illustrated in Fig. 4b. The collected emission at port C exhibits a maximum at a polarization angle of 90° and vanishes completely at 180°, which demonstrates that the TE-like EL signal is strongly polarized. The horizontal line in white corresponds to the polarization axis of the acquired spectra from the couplers shown in Fig. 5d.

The full-bandwidth enhancement-factor $F_{int}$ and coupling efficiency $\beta_{int}$ of the emitted EL signal are experimentally determined in the same fashion as with the incandescent NCG strip, integrated over the full wavelength of the signal. These figures of merits are derived for several cross-bar PhC devices with integrated sCNT as shown in Fig. 5c, f, respectively. The electrically-biased sCNT emits linearly polarized electroluminescence which is aligned with the tube axis and therefore also aligned with the TE mode of the cavity. We find $\beta_{int}$ as high as 99.5% and $F_{int}$ up to 188. It should be noted that all experimentally investigated devices showed coupling of EL from sCNT into the hybrid graphene-Si$_3$N$_4$ PhC cavities devices on-chip (Supplementary Fig. 10). The experimental results testify the stability and reproducibility of developed hybrid PhC devices with incorporated sCNT on a chip.

## Discussion

The heterogeneous integration of sCNT with cross-bar Si$_3$N$_4$ PhC devices provides a viable route towards electrically controlled nanoemitters with narrow-linewidth in the telecommunication band. The use of nanographene electrodes in particular enables seamless integration of electrically controllable devices with loss-sensitive photonic components, such as nanoscale cavities, and thus allows for realizing reconfigurable photonic circuits. Using electrically driven emitters removes the need to implement on-chip optical filtering which is challenging when high extinction ratio is required and increases the device footprint.

Our optimized fabrication protocol for the site-specific deposition of sCNT enables reproducibility for photonic applications in the telecom band. Using NCG electrodes provides the freedom to accurately position sCNT in a required region atop photonic device, such as PhC cavity, with negligible optical loss and degradation of the optical Quality factor. Thus, developed and demonstrated Hybrid graphene-Si$_3$N$_4$ device address key challenges in advanced photonic circuits which require both optical and electrical control on the nanoscale.

Besides applications in telecommunication, sCNT also provide promising characteristics for quantum photonic circuits. Cavity enhancement combined with low-loss electrical drive and back-gate configuration electrical control of the nanoemitter sCNTs may allow for realizing tailored single photon sources in a chip-scale framework. With recent advances in boosting the emission rates, our approach to site-selectively integrate sCNTs in telecommunication photonic cavities provides promising avenues for hybrid quantum photonic circuits.

## Methods

### Nanofabrication of hybrid PhC cavities with 3D couplers

The integrated hybrid cross-bar PhC devices are prepared from commercial 335 nm stoichiometric Si$_3$N$_4$ on 3.320 μm thick SiO$_2$ layer on top of 525 μm Si substrate. The synthesis of nanocrystalline graphene (NCG) on top of Si$_3$N$_4$ surface is performed by spin-coating photoresist (S1805 G2, Microposit) and graphitizing under high vacuum at high temperature. Details are described as the following recipe, resulting in a 5 nm thick layer. The photoresist solution is prepared by diluting to 1:4 ratio with Propylene Glycol Monomethyl Ether Acetate (PGMEA, Sigma-Aldrich). The prepared photoresist is spin-coated on the substrate followed by 110 °C annealing. The graphitization is obtained by further annealing (1000 °C for 10 h) in a high-temperature vacuum furnace ( < $10^{-6}$ mbar) equipped with a quartz glass tube. The thickness of the synthesized thin film is determined by atomic force microscopy in tapping mode.

Fabrication of hybrid nanophotonic circuits involves several steps of electron-beam lithography (e-beam) followed by reactive ion etching (RIE). In the first e-beam lithography step alignment markers were defined on the chip with positive photoresist PMMA. Gold (100 nm) with underneath chromium (7 nm) adhesion layer were deposited on chip by Electron Beam Physical Vapor Deposition followed by lift-off process in acetone. The main fabrication challenge is to pattern graphene and Si$_3$N$_4$ underneath with the condition that employed photoresist should be completely removed after reactive ion etching by wet-etching to avoid degradation of nanocrystalline graphene layer (which is not harmful for graphene layer) instead of standard routine - utilization oxygen plasma, since oxygen removes nanocrystalline graphene. We overcome it by the development and optimization recipe based on e-beam lithography patterning of negative tone photoresist ma-N 2403 with underneath sacrificial layer of PMMA. After electron-beam lithography exposure remaining not-exposed negative resist was removed in the developer MF-319. Further, the structures were fully dry-etched into the silicon nitride layer using an CHF$_3$/O$_2$ plasma, and both photoresists were lifted-off in acetone. In the next step designed graphene electrodes were defined in the similar spincoated sandwich of e-beam resist layers. After e-beam exposure and similar development process the non-covered graphene was removed in O$_2$ plasma, and again both photoresists were lifted-off in acetone. In the last e-beam lithography step contact pads were transferred into PMMA layer with further deposition of 7 nm Cr adhesion layer and 120 nm of Au, followed by lift-off in acetone. In the last fabrication step 3D coupling structures were realized on each device by optimized additive manufacturing using direct laser writing (DLW) with a Nanoscribe tool.

### sCNT preparation and deposition on chip

Single-walled sCNTs are synthesized by selective-catalyst chemical vapor deposition using CoSO$_4$/SiO$_2$ as a catalyst and CO as a carbon precursor. Details are described elsewhere[23]. The catalyst (200 mg) is loaded in a 1 in. tubular reactor and reduced under H$_2$ flow (1 bar,

50 sccm) while the reactor temperature is increased to 540 °C. Then, the reactor temperature is further increased to 780 °C with Ar purging (1 bar, 50 sccm). Afterward, the catalyst is exposed to CO (6 bar, 100 sccm) to initiate SWCNT growth for 1 h. Raw SWCNT soot is collected by dissolving the catalyst loaded with SWCNTs in NaOH (1 M) solution. The detailed preparation of toluene-based sCNT suspensions is described in our previous work[24]. For SWCNT suspensions 100 mg of the raw SWCNT soot and 100 mg of the polymer poly(9,9-di-n-dodecylfluorenyl-2,7-diyl) (PODOF) (Sigma-Aldrich) are mixed in 100 mL of toluene and subjected to a sonication treatment for 2 h by using a titanium sonotrode (Bandelin). During sonication, the suspension is placed in a water-circulation bath to aid cooling. After sonication, the suspension is then centrifuged for 2 h at 20,000 g. To generate the starting suspensions for size exclusion separation, the supernatant is concentrated to ~5 mL by evaporating ~95 mL of toluene. Semi-preparative size exclusion chromatography is performed using Toyo-pearl HW-75 resin (Tosoh Bioscience) filled into a glass column having a 16 mm inner diameter and 20 cm length. After application of 5 mL of SWCNT starting suspension to the gel, the sample is flowed through the gel under gravity, resulting in a flow rate of ~2 mL/min with toluene as eluent. Fractions are collected in ~4 mL portions. Individual sCNTs are bridged between multiple nanocrystalline graphene contact pairs by electric field-assisted dielectrophoresis[6]. The droplet of diluted suspension with individual tubes is placed onto the chip. A bias between 1 and 2.5 V at frequencies between 200 kHz and 1 MHz is applied between the common drain and the back-gate electrode using an Agilent 33250 function generator. To confirm the deposition of individual sCNTs, transport characteristics of the devices are carried out at ambient conditions in a probe station with TRIAX probes using an Agilent 4155 C semiconductor parameter analyzer.

### EL spectroscopy setup for at room and cryogenic temperature

The fabricated chip is mounted on a custom-made sample holder and placed into sample-in-vacuum optical cryostat system (MicrostatHiResII, Oxford). The devices are electrically connected onto palladium pads attached to the sample holder by means of wire bonding. Samples are vacuum annealed ($<10^{-6}$ mbar) at 70–120 °C to improve contact conductivity while avoiding degradation of 3D coupling structures. Without breaking the vacuum, subsequent measurements are performed. Devices are electrically driven by an Agilent 4155B semiconductor parameter analyzer. Constant-current biasing is applied on source and drain electrodes through separate source-measurement units (SMU) and a third SMU is used for gate voltage while the drain electrode is set as reference. The optical access to the sample is through a 0.5 mm thick quartz window with an Olympus LMPLN 10XIR objective mounted on a customized Zeiss Axiotech Vario microscope. The emitted light is focused with an off-axis parabolic mirror (MPD149-P01, Thorlabs) into an imaging spectrograph (Acton SP-2360, Princeton Instruments) and dispersed via a 85 G/mm, 1.35 μm blazed grating onto an InGaAs photodiode linear array (PyLoN-IR, Princeton Instruments) with 1024 pixels, sensitive from 950 to 1610 nm. The xy position of the sample is controlled with sub-μm precision by a motorized scanning stage (8MTF, Standa), and the z-axis working distance is adjusted by a high-precision objective positioner (P-721 PIFOC/E-665 piezo scanner, Physics Instruments), which allowed precise and stable positioning of the emitter. All electroluminescence spectra are corrected by the relative spectral sensitivity of the setup and the measured line width was limited by the ~2 nm spectral resolution.

## Data availability

Relevant data supporting the key findings of this study are available within the article and the Supplementary Information file. All raw data generated during the current study are available from the corresponding authors upon request.

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

## Acknowledgements
A.P.O. thank Alexander Eich for help with acquiring He microscope image of 3D coupler. M.L., F.H., and R.K. acknowledge support from the Helmholtz Research Programs Natural, Artificial and Cognitive Information Processing (NACIP), and by the Karlsruhe Nano Micro Facility (KNMF). F.P., R.K., and WP acknowledges support from the Volkswagen Foundation. W.H.P.P. acknowledges support by the European Research Council (ERC, PINQS Project). H.G. acknowledges the Studienstiftung des deutschen Volkes for financial support. We thank Jochen Stuhrmann for assistance with the graphical illustrations. This work was supported by the European Union's Horizon 2020 research and innovation programme (grant no. 101017237, PHOENICS Project) and the European Union's Innovation Council Pathfinder programme (grant no. 101046878, HYBRAIN Project). We acknowledge funding support by the Deutsche Forschungsgemeinschaft (DFG, German Research Foundation) under Germany´s Excellence Strategy EXC 2181/1 – 390900948 (the Heidelberg STRUCTURES Excellence Cluster), the Excellence Cluster 3D Matter Made to Order (EXC-2082/1—390761711) and CRC 1459 "Intelligent matter".

## Author contributions
W.H.P.P., R.K., F.P., and A.P.O. conceived the experiment. W.H.P.P. and R.K. supervised the project. A.P.O. designed, simulated, optimized, fabricated hybrid PhC cavity devices with nanocrystalline graphene electrodes. A.P.O. performed 3D FDTD simulations and optimized PhC cavity. F.P. and M.-K.L. developed the measurement electroluminescent spectroscopy setup. A.P.O., M.-K.L. and F.P. performed experimental electroluminescent measurements of sCNT coupled to hybrid PhC cavity device. F.P. and M.-K.L. deposited sCNT on the hybrid PhC cavity devices. H.G. optimized and fabricated 3D couplers. F.B. prepared Python codes for electron beam lithography. M.-K.L. and S.K. prepared nanocrystalline graphene on the chips. The nanotube raw material was provided by L.W. and Y.C., F.H purified and length fractionated nanotubes. A.P.O. and W.H.P.P. prepared the article with input from M.-K.L., F.P. and R.K.

## Funding

## Competing interests
The authors declare no competing interests.
