## [Peer Review File · Nature Communications]

An electroluminescent and tunable cavity-enhanced carbon-nanotube-emitter in the telecom bandReviewers' comments:

Reviewer #1 (Remarks to the Author):

I read the manuscript entitled "A cavity enhanced and tunable electroluminescent carbon nanotube emitter in the telecom band.". Scalable nanoscale light source integration to the photonic circuit is an essential technology for realizing the integrated photonic circuits and information process. The authors demonstrated the semiconductor Carbon Nanotube (sCNT) emitter to the hybrid photonic crystal structure with nanocrystalline graphene (NCG) for electrode materials. In this manuscript, they presented that the telecom band emission from sCNT and NCG is low-loss material to electrical contact. The authors claimed that demonstrated methods of CNT dielectrophoresis enable the scalable fabrication of the light source integrated on-chip photonic circuits. The authors did not present the sCNT emitter array for integrated photonic circuits in the manuscript. Furthermore, essential on-chip photonic circuit light sources must show high-speed modulation and high modulation depth. In this manuscript, the authors do not include these experiments results. Compared to the previous works [Nature Photon 10, 420 (2016), Nat Commun 10, 109 (2019)], I can't find any significant improvement or breakthrough results. So I don't recommend the publications to Nature Communications.

1.The authors claimed that dielectrophoresis provides a stable deposition for sCNT and allows for deterministic placement between electrodes. However, the authors do not show any SEM image of the sCNT placed onto the photonic crystal and NCG electrodes. I am also wondering if the authors can put the individual sCNT or bundles of sCNTs onto the photonic crystal and NCG electrodes.

2.The authors should include the IV characteristics of sCNT and NCG.

3.NCG-based thermal radiation shows an on-resonance coupling efficiency of 99.1% at 300 K and 98.7% at 77K. Why is room temperature on-resonance coupling efficiency larger than a low-temperature coupling efficiency?

4.What is the maximum electron temperature of NCG and melting temperature?

5.In Fig. 5b, the Off-state mean transistor is off-state, whereas EL emission is on-stage. Due to the off-state and EL emission on-state at the same regime led to a misunderstanding of the paragraph and confusion.

Reviewer #2 (Remarks to the Author):

This manuscript from Anna P. Ovvyan et al reports on the electroluminescence of a single semiconducting (9,8) carbon nanotube coupled to a photonic cavity. The main noteworthy result of this manuscript is the deterministic placement of a single (9,8) carbon nanotube inside a photonic crystal cavity using dielectrophoresis. The electrodes used for the nanotube precise deposition are further used to electrically drive the nanotube luminescence.

The authors chose a cross-bar photonic crystal cavity design. The main advantage with this approach is the direct coupling with photonic waveguide, which allow further integration with photonic circuit. To overcome the optical loss inherent in using metal electrode near the photonic crystal cavity, the authors use graphene electrodes instead, which are demonstrated to be low loss. After a characterization of the cavity using the black-body emission from a graphene strip inserted into the cavity, the authors successfully demonstrate the electroluminescence from the (9,8) nanotube, and its electrical driving.

This is an interesting paper with a strong demonstration of proper electroluminescence from semiconducting carbon nanotube coupled into an easily integrable photonic cavity. However, a similar work was already demonstrated by the same group ("cavity-enhanced light emission from

electrically driven carbon nanotubes" Nature photonics, doi:10.1038/NPHOTON.2016.70), where light emission from carbon nanotube was coupled into a suspended photonic waveguide (nanobeam cavity). The improvement of this work compared to the previous work mainly reside in the clever deposition method of the nanotube, dielectrophoresis using low loss graphene electrode, and the cross-bar cavity design which allow further integration into photonic circuitry.

The use of graphene black body emission by itself was already published in "thermal radiation control from hot graphene electrons coupled to a photonic crystal nanocavity", Nature communications, doi:10.138/s41467-018-08047-3.

While this paper is properly documented, I don't feel that there is new insight that justify publication in a broad audience journal such as Nature communications. On the other hand, the technological improvements are interesting and deserve to be published in a more specialized journal.

Should this manuscript be resubmitted elsewhere, I suggest that the authors correct the typo and in particular the errors with the references. In addition, some statistics about the working devices should be provided, such as the ratio of working devices over the fabricated devices, and if possible, the device variability. For example, the device 2 in Fig5 display a significantly lower source-drain current than devices 1 and 3. The authors should comment theses points.

Reviewer #3 (Remarks to the Author):

Anna P. Ovyvan et al report a paper on "A cavity enhanced and tunable electroluminescent carbon nanotube emitter in the telecom band". The authors propose an original device configuration where a single CNT is coupled to nanographene electrodes for the carrier injection and to a photonic cavity for the extraction of photons. This topic is very interesting and the results are convincing. Nevertheless, I have few concerns before I can recommend this paper for publication in Nature Communications:

1- The authors mention several times in the main text the Purcell effect. Nevertheless, there is no experimental proof that the Purcell regime is achieve here. This takes nothing away from the interest of the results reported in the paper. But, the author should stay with enhancement factors, meaning enhanced collection of photons due to the redirection in the optical mode of the cavity.

2- On figure 5d), the EL spectra in different experimental conditions are plotted. In particular, if I understand, the green curve is supposed to show the one of the CNT detected in the confocal configuration (so not coupled to the cavity). Nevertheless, nothing is visible on the figure. So, is there any EL signal of the CNT?

3- The authors, show also EL from the nanographene electrodes. If I am right, they observe it at much higher current than for the EL of the CNT. Is it right? If yes, I suggest that they insist on this point in the text.

4- In a previous work, the authors demonstrated the single photon emission of CNT with an electrical injection. Did the authors tried to perform correlation measurements in their devices? This would greatly enhance the interest of the paper.

5- Finally, the authors mention the possibility of scalability of their technics. Here, "only" 3 devices are shown. I would suggest that the authors discuss the scalability at this step of their project. How is efficient the placement of the CNT? When a CNT is well placed on the cavity, how many have the good exciton energy to be in resonance? Etc...

Response to the reviewer comments

We would like to thank the reviewers for their helpful comments on now to improve the manuscript. We have responded to the recommendations in full and have adjusted the main text and the supplementary materials accordingly. In the following, we address the reviewer's comments and questions one by one summarized below.

We organize Response to the Review comments in the following way. The comments of the Reviewer are in black, our (authors) responses are in violet, citations/quotes from our manuscript and SI are in red.

Reviewer #1 (Remarks to the Author):

I read the manuscript entitled “A cavity enhanced and tunable electroluminescent carbon nanotube emitter in the telecom band.”. Scalable nanoscale light source integration to the photonic circuit is an essential technology for realizing the integrated photonic circuits and information process. The authors demonstrated the semiconductor Carbon Nanotube (sCNT) emitter to the hybrid photonic crystal structure with nanocrystalline graphene (NCG) for electrode materials. In this manuscript, they presented that the telecom band emission from sCNT and NCG is low-loss material to electrical contact. The authors claimed that demonstrated methods of CNT dielectrophoresis enable the scalable fabrication of the light source integrated on-chip photonic circuits. The authors did not present the sCNT emitter array for integrated photonic circuits in the manuscript. Furthermore, essential on-chip photonic circuit light sources must show high-speed modulation and high modulation depth. In this manuscript, the authors do not include these experiments results.

Compared to the previous works [Nature Photon 10, 420 (2016) - Valentin, Nat Commun 10, 109 (2019) – Englund graphene], I can't find any significant improvement or breakthrough results. So I don't recommend the publications to Nature Communications.

Our response: We thank the referee for her/his assessment of our work, but do disagree with the overall remark on the novelty of our work. Indeed, the key difference in this work is the demonstration of electroluminescent performance, which is markedly different from the incandescent results in the work of Fütterling et al.

However, we performed new experimental measurements to address the concerns of the Reviewer (the experiments required certain amount of time to be accomplished), we added the results in the new Supplementary sections 6-10.

1. The Reviewer mentioned that “The authors did not present the sCNT emitter array for integrated photonic circuits in the manuscript.”

Our response: We thank the Reviewer for the comment. In order to reply to this comment, we **performed new measurements** and demonstrated further electrical control of the intensity of enhanced electroluminescence from a single sCNT coupled to a hybrid nanographene-PhC cavity device. The dynamic control with close to 100% on-off ratio (depth) was obtained by change of the back-gate voltage which produces the electric field between the sCNT and the Si substrate. We summarized the results in the **new Supplementary Section 6**. This confirms the fact that our novel method is scalable to the arrays of devices, taking into account also the measurements which are shown in Fig. 5 c-f.

We added the new **Supplementary Section 6** (see below):

[**6. Dynamic control of EL from cavity-integrated sCNT**

As a proof of principle of our versatile approach of full electrical control of single sCNT nanoemitter integrated in custom-designed low-loss nanographene-photonic environment, we demonstrate in this section the measurements of the other hybrid device, where sCNT EL coupled to nanographene-PhC cavity is dynamically operated by regulation of back-gate voltage.

Electrically-biased sCNT integrated in the cross-bar PhC cavity with simultaneously applied back-gate voltage ($U_g = -25V$) leads to an enhanced excitonic EL emission coupled into the odd resonance modes of the cavity in agreement with LDOS-spatial maps (Fig. 4a, S4a). The measured spectrum of light outcoupled from one of the ends (coupler C in Fig. 1c) of the investigated hybrid device is shown in Fig. S6a (red curve). This is obtained when sCNT is charged neutral, which corresponds to the EL *switched-on* state. Changing the gate voltage from $-25V$ to $+30V$ leads to decrease of intensity of the excitonic EL emission which is further coupled to resonance modes and outcoupled via coupler C (light and dark green curves in Fig. S6a). Switching the gate voltage to $U_g = +30V$ leads to complete suppression of the excitonic EL emission, corresponding to EL *switched-off* state (blue curve in Fig. S6a). Thus, we obtain dynamic control of sCNT enhanced EL emission with close to 100% on-off ratio (depth) via active electrical operation of back-gate voltage.

Figure S6. Experimental dynamic control of EL from cavity-integrated sCNT. *a)* The spectra of EL sCNT coupled to odd resonance modes at 1419.5nm, 1434.4nm, 1452.9nm, acquired from coupler C of NCG-Si₃N₄ PhC cross-bar device gated with corresponding voltage U_g . At $U_g = -25V$ – EL is in switched-on state (red curve) and at $U_g = +30V$ – EL is switched-off (blue curve), sCNT biasing current is constant $I_{sd} = 30nA$. Cross-bar PhC cavity consists of $N = 45$ segments in each Bragg mirror with a lattice period of $a = 457nm$. The resonance modes are labelled. *b)* $U_{sd} - U_g$ curve acquired at constant sCNT biasing current $I_{sd} = 30nA$. Forward and backward sweep traces are shown. The transport data shows negligible hysteresis between the forward and backward sweeps due to the cryogenic environment (77 K). The yellow area corresponds to the regime, in which the excitonic EL emission is in the switched-on state. All data recorded at 77K.]

To refer to the Supplementary Section 6, we added the following sentence in the main text of the manuscript:

[As a proof of principle of our versatile approach of full electrical control of EL sCNT, we demonstrate in the Supplementary Section 6 experimental measurements of the other hybrid device where the enhanced EL-sCNT coupled to a low-loss nanographene-photonic environment is dynamically operated via back-gate voltage regulation.]

2. The Reviewer mentioned that “Furthermore, essential on-chip photonic circuit light sources must show high-speed modulation and high modulation depth.”

Our response: We thank the Reviewer for the comment. Our novel demonstration of dynamic control of the EL intensity of the sCNT with close to 100% on-off depth (ratio) is realized via operation of the back-gate voltage which creates the electric field between the sCNT and the Si substrate.

The cut-off frequency of sCNT modulation in this field-effect configuration with applied back-gate voltage is determined by the time constant of the electrically equivalent RC-circuit. Notably, the sCNT excitonic lifetime is on the order of tenths of picoseconds, which is reduced even further by placing the sCNT in a PhC cavity due to EL enhancement. Thus, it does not limit the cut-off frequency. Unfortunately, we were limited by our vacuum optical cryostat measurement setup in performing high-speed modulation measurements. Our configuration based on single sCNT emitter (owing to high carrier mobility and small intrinsic capacitance) well-aligned in between nanocrystalline graphene electrodes on the cavity promise operation frequencies above 100 GHz via dynamic electrical control of back-gate voltage. This could potentially be measured with suitable RF access, which in our case is not available in the low-temperature measurement station.

We believe that our realized and demonstrated full control of enhanced single sCNT EL performed via change of back-gate voltage with high on-off ratio in the telecommunication window via a designed ultra-low-loss electrical contacting scheme based on NCG electrodes ensures an elegant way to electrically control photonic devices without inducing absorption or scattering losses and provides promising avenues for hybrid classical and quantum photonic circuits.

3. The Reviewer mentioned that “Compared to the previous works [Nature Photon 10, 420 (2016), Nat Commun 10, 109 (2019) – Englund graphene], I can’t find any significant improvement or breakthrough results. So I don’t recommend the publications to Nature Communications.”

Our response: We politely disagree with this point.

A) Below we highlight the novelty of our work and show the fundamental differences of our work in comparison with mentioned above work of F. Pyatkov *et al.* [Nat. Photon. 10, 420–427 (2016)] [7].

First, F. Pyatkov *et al.* [7] electrically excited **broadband incandescent emission (thermal radiation)** of a biased **metallic CNT**, which was coupled into the nanobeam cavity. In contrast, we integrated **semiconducting CNT** and excited fundamentally different nature of light - **electroluminescence (EL)** of **sCNT**, and coupled it into Photonic Crystal Cavity. Importantly, we applied three orders of magnitude lower biasing current (nA range) to generate EL sCNT in comparison with incandescent emission in [7].

Second, in contrast with the work [7] we **for the first time experimentally obtained full dynamic control** of the intensity of the **enhanced EL** of a **single sCNT** (Fig. 1d) with 100% on-off ratio by change of back-gate voltage, which produces electric field between sCNT and Si substrate (without dissipation of energy). The above-mentioned control is demonstrated in Fig. 5d,e and in **the newly added Fig. S6. This is direct confirmation of the electroluminescence nature of the coupled sCNT emission.**

Third, we developed a novel, ultra-low-loss electrical contacting scheme based on nanocrystalline graphene electrodes which provides an elegant way to electrically control photonic devices without inducing absorption or scattering losses (as illustrated in Fig. 2) in comparison with metal electrodes utilized in the work F. Pyatkov *et al.* [7]. This allows to place electrodes directly on the cavity for the accurate alignment of sCNT in a required region, namely in the maximum of electric field of the resonance mode (Fig. 4a and Fig. S4a) without degradation of the cavity Q-factor as illustrated in Fig. S2.

Fourth, importantly, our sCNT-emitter operates in the wide telecommunication window (1400-1550nm) and thus in a technologically viable wavelength range that has not been in reach before, while in the work F. Pyatkov *et al.* [7] working wavelength region is $\lambda=900-1000\text{nm}$.

Fifth, in contrast with work [7] where F. Pyatkov *et al.* utilized fragile and in terms of nanofabrication a freestanding and more complicated (suspended) nanobeam cavity, we developed stable and thus easy to fabricate tailor-made cross-bar PhC cavities on an SiO₂ substrate (Fig. 1d-e and Fig. S1a), leading to experimental high enhancement factor ($F_{int}=188$) and coupling efficiency ($\beta_{int} = 99.5\%$) of EL coupled into resonance modes (Fig 5c,f). This enhancement is more than 30x stronger compared to [7] and substantially more efficient (compared to 30% in [7]).

Sixth, in contrast with work [7], where F. Pyatkov *et al.* utilized standard grating couplers, our developed hybrid photonic devices are equipped with broadband, highly efficient 3D polymer couplers (Fig. 1b) for convenient out of plane optical read out of the enhanced EL signal in a wide telecommunication band. Thus, our hybrid photonic device can be further operated as a broadband emitter covering the wavelength window from 1400 to 1600 nm.

To highlight the differences of our work in comparison with F. Pyatkov *et al.* [Nat. Photon. 10, 420–427 (2016)] [7] we summarized the main above-mentioned points in the table below.

	Our work demonstrated in this Manuscript	F. Pyatkov et al. [Nat. Photon. 10, 420–427 (2016)] [7]
Type of integrated CNT	Semiconducting CNT (sCNT)	(metallic) CNT
CNT detected emission light	Electroluminescent (excitonic) emission (EL)	Incandescent emission
Emission (working) wavelength range	Telecommunication band (1400-1500nm)	Near-visible range 900-1000nm
Control of enhanced CNT emission	First time demonstrated full electrical control of enhanced electroluminescent emission intensity of sCNT by change of back-gate voltage, which produces electric field	Electrical control of incandescent emission via change of bias of CNT. gate was not provided.

	between sCNT and Si substrate (field-effect configuration)	
Electrical contacting scheme	Novel, ultra-low optical loss electrical contacting scheme based on nanocrystalline graphene electrodes which does not degrade Q-factor of PhC cavity	Absorptive metal (gold) electrodes which suppressed resonance modes of the cavity
Experimentally determined enhancement of the CNT emission	$F_{int}=188$ (Enhanced EL emission)	$F_{on_resonance}=5$ (Enhanced incandescent emission)
Experimentally determined coupling efficiency β_{int} of the CNT emission	$\beta_{int}=99.5\%$ (Coupled EL emission)	$\beta_{int}=30\%$ (Coupled incandescent emission)
Type of the cavity	Not-freestanding and easy in fabrication cross-bar PhC cavity on SiO ₂ substrate	Fragile Freestanding (suspended) nanobeam PhC cavity
Couplers for out of plane read-out optical enhanced emission signal	High-efficiency broadband 3D polymer couplers covering wide telecommunication range [18]	Grating couplers

In order to highlight the fundamental difference of our work in contrast with F. Pyatkov *et al.* [Nat. Photon. 10, 420–427 (2016)] [7] we added the following sentences to the main text of the manuscript (see below).

[The applied low driving bias-current (tens of nanoamperes) to the sCNT further proves the electroluminescent nature of the emitted light, in contrast to previous work by Pyatkov *et al.* [7] where three orders of magnitude higher biasing current was utilized to generate incandescent emission of CNTs. We achieve an *on-off* ratio close to 100%.]

[Importantly, the utilized biasing current of the sCNT is three-four orders of magnitude lower in comparison with the nanocrystalline graphene strip thermal emitter described in the previous paragraph, as well as in comparison with the incandescent CNT realized in the work of Pyatkov *et al.* [7]. This again confirms the EL nature of sCNT emission in contrast with incandescent NCG emission.]

In order to emphasize and clarify the novelty of our work, we **modified and added the following paragraphs** in the Abstract, Introduction and Conclusion of the main manuscript text (see below).

[We demonstrate enhanced spectral line shaping of the EL sCNT emission. By back-gating the sCNT-nanoemitter we achieve full on-chip electrical dynamic control of the EL sCNT emission with high on-off ratio and strong enhancement in the telecommunication band. Using nanographene as a low-loss material to electrically contact sCNT emitters directly within a photonic crystal cavity enables realization highly

efficient EL coupling without compromising the optical quality of the cavity. Our integrated hybrid approach allows to redirect the enhanced EL signal propagating in photonic waveguide to 3D polymer couplers for convenient out of plane optical read-out. Our versatile procedure of site-selective placement of electroluminescent sCNT emitters in custom-designed low-loss nanographene-photonic environments with high fidelity and reproducibility paves the way for controllable integrated photonic circuits on-chip.]

[We demonstrate enhancement of the EL emission of sCNTs by efficient coupling into a hybrid nanographene-PhC cavity device. Importantly, we obtain full dynamic control of the intensity of the enhanced EL sCNT by active electrical operation of the back-gate voltage, which in fact proves the electroluminescent nature of the emitted sCNT light.]

[The electrically controllable hybrid NCG-Si₃N₄ photonic circuits with deterministic placement of sCNTs in the cavity region provide a scalable and reproducible platform which meets the requirements of integrated photonic circuits for classical and quantum applications.]

[As a proof of principle of our versatile approach of full electrical control of an EL-sCNT, we demonstrate in the Supplementary Section 6 the experimental measurements of the other hybrid device where the enhanced EL-sCNT (which is coupled to a low-loss nanographene-photonic environment) is dynamically operated via back-gate voltage regulation.]

[Notably, the presence of the EL signal at the end of the PhC cavity (red curve in Fig. 5d) and the simultaneous absence of the free-space EL signal detected at the position of the investigated coupled sCNT (green curve in Fig. 5d) in the switched-on state ($U_g=20V$) of the demonstrated device testifies successful coupling of EL into the fundamental resonance mode of the PhC cavity.]

[Cavity enhancement combined with low-loss electrical drive and back-gate configuration electrical control of the nanoemitter sCNTs may allow for realizing tailored single photon sources in a chip-scale framework. With recent advances in boosting the emission rates, our approach to site-selectively integrate sCNTs in telecommunication photonic cavities provides promising avenues for hybrid quantum photonic circuits.]

B) Furthermore, we highlight below the differences of our work in comparison with Shiue *et al.* [*Nat. Commun.* **10, 109 (2019)] [21].**

We incorporated a thermal nanocrystalline graphene strip nanoemitter in a predetermined position within a photonic device which allows us to probe the LDOS factor and provides optimal coupling of emitted light into cavity modes at cryogenic and room temperature. We would like to highlight that in contrast with the work of Shiue *et al.* [21], we demonstrated a strip-nanoemitter which can be integrated on nanophotonic devices **for high-resolution LDOS sensing owing to nano-dimensions and importantly without degradation of quality factor of the cavity** (Fig. 2, Fig. 3 and Fig S2, S3) at cryogenic and room temperature. Furthermore, the emitter can be easily removed by oxygen plasma (while protecting the rest of the circuit with photoresist, for example). We also note, that in the work of Shiue *et al.* [21] ~ three-four-times degradation (drop) of Q-factor of 2D PhC cavity was found after graphene deposition on top as well as red-shift of the resonance modes. In our case, our approach maintains high quality factors due to negligible influence of the NCG electrodes on the cavity mode.

To highlight the difference of our work in comparison with Shiue *et al.* [21] we added the following sentences in the main text of the manuscript (see below).

[Such a thermal strip-nanoemitter incorporated in a predetermined position within a photonic device allows to probe the LDOS factor and provides optimal coupling of emitted light in cavity modes at cryogenic and room temperature. The nanocrystalline graphene strip nanoemitter can be removed by oxygen plasma (while protecting the rest of the circuit with photoresist, for example) and thus makes it a promising candidate for optical high-resolution LDOS sensing at room and cryogenic temperature. Importantly, NCG strip-nanoemitter doesn't degrade quality factor of the cavity as shown in Fig. S2c in contrast with the work of Shiue *et al.* [21], where it is demonstrated a micro black-body radiator for optical communication application.]

4. The authors claimed that dielectrophoresis provides a stable deposition for sCNT and allows for deterministic placement between electrodes. However, the authors do not show any SEM image of the sCNT placed onto the photonic crystal and NCG electrodes. I am also wondering if the authors can put the individual sCNT or bundles of sCNTs onto the photonic crystal and NCG electrodes.

Our response: We thank the Reviewer for this comment. In order to reply to this question, we took **new SEM image** of the investigated single sCNT deposited between NCG electrodes in the cavity region (by site-selective dielectrophoresis) of the experimentally measured hybrid device. **We added this SEM image in Fig. 1d with the inset showing a close-up image of an individual sCNT in the cavity.** We dynamically control EL of this individual sCNT which is demonstrated in Fig. 5b,d,e.

Modified Figure 1 in the Manuscript with the new SEM image of single sCNT in the cavity region in (d) is shown below.

[

Figure 1. Electrically controlled cross-bar PhC cavity with integrated sCNT emitter. **a)** Schematic of the hybrid device. The sCNT is positioned between nanocrystalline graphene electrodes placed on top of the cavity crossing. The nanographene electrodes are used for addressing the sCNT and are connected to metallic leads for current injection. The electroluminescent emission is coupled out from both ends of the PhC cavity. **b)** Helium Ion microscopy image of a broadband total reflection 3D coupler connected to the ends of the nanophotonic waveguides. **c)** Optical micrograph of the fabricated device with 3D output couplers and cavity region marked by the zoom-box. **d)** Scanning electron microscope image of PhC cavity with single sCNT integrated between nanocrystalline graphene electrodes. The inset shows a close-up image of an individual sCNT in the cavity. **e)** Scanning electron microscope image of the hybrid device in the cavity region.]

We also added the following sentence in the Manuscript text:

[Controllable integration of single sCNT in the required location is achieved by self-alignment between electrodes through the induced electrostatic field during dielectrophoresis. Thus, owing to the design, a single sCNT is placed in the optimal position in the center of the cavity as shown in Fig. 1d (see inset)..]

Thus, dielectrophoresis provides a stable and deterministic deposition method for single sCNTs between NCG electrodes on the hybrid devices. Successful deposition is obtained with optimized parameters for the applied electric field as well as an optimized concentration of sCNT in diluted suspension. Furthermore, to confirm the deposition of individual sCNTs, transport characteristics of the devices were carried out at ambient conditions in a probe station with TRIAX probes using an Agilent 4155C semiconductor parameter analyzer. The detailed recipe of sCNT deposition and parameters of Dielectrophoresis are described in the Methods and Fabrication section.

5. The authors should include the IV characteristics of sCNT and NCG.

Our response: We thank the Reviewer for this suggestion. In order to reply to this question, we performed **new measurements** of I-V curves of electrically-biased single sCNT (9,8) and nanocrystalline graphene strip, we added the results in the **new Supplementary Section 7**.

We added the new **Supplementary Section 7** (see below):

[7. I-V curves of sCNT and NCG strip

I-V curves of sCNT (9,8) integrated in between NCG electrodes in the EL *switched-on* (blue curve) and EL *switched-off* (red curve) states are shown in Fig. S7a. I-V curve of investigated nanocrystalline graphene strip with a narrow junction between NCG electrodes is demonstrated in Fig. S7b. Linear increase of source-drain current with applied voltage proves good Ohmic contact indicating low resistance in case of NCG strip (Fig. S7b), while in case of sCNT the behavior of I-V curve is non-linear (Fig. S7a), which is the signature of non-ohmic contact, implying a higher barrier between the polymer-wrapped (9,8) sCNT and NCG nano-electrodes.

Figure S7. Measured I-V curves of sCNT (9,8) (a) and NCG nano-strip (b). Thickness of NCG – 5nm.]

6. NCG-based thermal radiation shows an on-resonance coupling efficiency of 99.1% at 300 K and 98.7% at 77K. Why is room temperature on-resonance coupling efficiency larger than a low-temperature coupling efficiency?

Our response: We thank the Reviewer for this comment. Firstly, we assume that after successful deposition of sCNT between electrodes on the devices, the residuals of the suspension remain (randomly distributed) on the devices, and in particular on the devices with NCG-strips. [The detailed information regarding sCNT deposition is provided in the Methods section]. After performing measurements at 77K and heating up to 300K, the residuals of the suspension evaporated, which leads to a decrease of the effective refractive index of the resonance modes of the PhC cavity, resulting in a blue shift of resonance wavelength of these modes. This is evident by comparing the spectra detected at the 3D couplers, shown in Fig. 3d (77K) and Fig.S3b (300K).

Secondly, the rise of the ambient temperature (77K→300K) leads to an increase of NCG resistivity, resulting in larger NCG applied electrical power which is necessary to keep the biased source-drain current constant at 120uA in both experiments at 77K and 300K. This leads to higher intensity of thermal radiation from NCG. This is evident in the higher emission intensity detected from the 3d couplers and the NCG strip at 300K, shown in Fig. S3b in comparison with the spectrum measured at 77K (Fig. 3d).

Thirdly, taking into account both above-mentioned effects, and in particular the random distribution of the residuals of the suspension on the chip at 77K, this led after evaporation of the residuals at 300K to an increase of the on-resonance emission intensity outcoupled from couplers by a factor of 10.7, 4 and 3.8, in case of the I, III, V resonance modes, respectively in comparison with the measurements at 77K. While, on-resonance the free-space intensity extracted from the NCG strip increases by a factor of 3.36, 2.89 and 3.62 in case of the I, III, V resonance modes, respectively in comparison with the measurements at 77K. **This explains the behavior of β , which is questioned in the comment of the Reviewer. Namely, this led to a slightly higher on-resonance enhancement factors (F_I and F_{III}) and coupling efficiency (β_I and β_{III}) determined at room temperature in comparison with the values extracted at 77K.**

7. What is the maximum electron temperature of NCG and melting temperature?

Our response: We thank the Reviewer for the comment.

In order to reply on this question, we performed **new measurements and added the results in the new Supplementary Section 8**. Namely, we experimentally measured the free-space incandescent emission spectrum of an electrically-biased nanocrystalline graphene strip placed on a Si_3N_4 waveguide (similar device design as shown in Fig. 3c, but the NCG strip is placed on waveguide instead of cavity). The detected monotonic thermal spectrum (red curve) is shown in Fig. S8 (Supplementary Section 8).

Importantly, in our experiments (Fig. 3 and Fig. S8), the utilized NCG strip (thickness 5 nm) consists of ~ 15 layers of single-layer graphene, and it is placed on a Si_3N_4 cavity or waveguide (the NCG strip is not suspended). Both of these facts allow us to assume that the NCG strip infrared emission represents an electronic temperature, since there are no significant non-equilibrium phonon distribution present [Kim Y. *et al.* Nature Nanotech 10, 676–681 (2015)], [Freitag M. *et al.* Nature Nanotech 5, 497–501 (2010)].

Thus, we fit the incandescent emission of the biased NCG strip at an applied electrical power of 2.87 mW (red curve in Fig. S8a) with a grey-body theory (Planck's law, modified by an emissivity) (1), and determine the electron temperature of the NCG – T_e of ~ 1000K, as indicated by the dashed blue curve in Fig. S8.

$$I(\lambda, T) = \varepsilon * \frac{2\pi hc^2}{\lambda^5} * \frac{1}{\exp\left(\frac{hc}{\lambda k_B T} - 1\right)}, \quad (1)$$

where we used $I(\lambda, T)$ – spectral energy density of thermal radiation from the NCG strip, h – Planck constant $6.626 * 10^{-34}$ ($J * sec$), k_B – Boltzmann const $1.380 * 10^{-23}$ ($\frac{J}{K}$), T – electron temperature (K) of NCG strip, λ – wavelength of the emitted photons, c – speed of light, ε – emissivity of the NCG strip ($\varepsilon \approx 0.25$) [S-E Zhu *et al.* 2014 *EPL* 108 17007], [Freitag M. *et al.* Nature Nanotech **5**, 497–501 (2010)].

We also comment here regarding the maximum temperature of NCG: Ganz *et al.* (Phys. Chem. Chem. Phys., 2017, 19, 3756) have reported that the lattice of freestanding monolayer graphene could well maintain a temperature of 4000K and start melting at 5000K. Recent work from Gamboa-Suárez *et al.* (Carbon Trends 9, 100197, 2022) has shown that the mechanical properties and structure of graphene and NCG are preserved at 5000K and 4000K, respectively. An electron temperature of NCG above 1500K has been demonstrated by A. Riaz *et al.* (Nanotechnology, 2015, 26, 325202). The highest electron temperature of suspended mechanically exfoliated few-layer and multilayer graphene reported so far is around 2850K (Y.D. Kim *et al.*, Nature Nanotechnology 10, 676–681, 2015).

We have added the new Supplementary Section 8 (see below):

[8. Telecom NCG-based incandescent nanoemitter electron temperature

We experimentally measured free-space incandescent emission spectrum of electrically-biased nanocrystalline graphene strip placed on Si₃N₄ waveguide (similar device design as shown in Fig. 3c, but NCG strip is placed on waveguide instead of cavity). The detected monotonic thermal spectrum (red curve) is shown in Fig. S8.

Importantly, in our experiments (Fig. 3 and Fig. S8), utilized NCG strip (thickness 5 nm) consists ~ 15 layers of single-layer graphene, and it is placed on Si₃N₄ cavity or waveguide (NCG strip is not suspended). Both of these facts allow us to assume that NCG strip infrared emission represents an electronic temperature, since there are no significant non-equilibrium phonon distribution exists [3] [4].

Thus, we fit incandescent emission of biased NCG strip at applied electrical power 2.87 mW (red curve in Fig. S8a) with a grey-body theory (Planck's law, modified by an emissivity) (1), and determine electron temperature of NCG – T_e of ~ 1000K, as indicated by dashed blue curve in Fig. S8.

$$I(\lambda, T) = \varepsilon * \frac{2\pi hc^2}{\lambda^5} * \frac{1}{\exp\left(\frac{hc}{\lambda k_B T} - 1\right)}, \quad (1)$$

where $I(\lambda, T)$ – spectral energy density of thermal radiation from NCG strip, h – Planck constant $6.626 * 10^{-34}$ ($J * sec$), k_B – Boltzmann const $1.380 * 10^{-23}$ ($\frac{J}{K}$), T – electron temperature (K) of NCG strip, λ – wavelength of the emitted photons, c – speed of light, ε – emissivity of NCG strip ($\varepsilon \approx 0.25$) [4] [5].

Figure S8. Emission spectrum of NCG strip nano-emitter on Si_3N_4 waveguide. *a)* Recorded free-space thermal emission spectrum of electrically-biased NCG strip using 70 μA (source-drain voltage 41V). The spectrum from the NCG strip was acquired with the polarizer parallel to TE mode of the waveguide. The waist of NCG strip is 250 nm in width, thickness – 5 nm. Drop of the detected intensity at $\lambda=1600\text{nm}$ is due to cut-off of sensitivity of utilized InGaAs photodiode linear array. Blue dashed curve: grey-body fit spectrum at extracted temperature $T=1000\text{K}$ according to equation (1), where full fit spectrum is shown in *b)*.]

To refer to the Supplementary Section 8, we added the following sentence in the main text of the manuscript:

[Notably, the determined electron temperature of the NCG-based strip-emitter on Si_3N_4 waveguide at an applied electrical power of 2.87 mW (source-drain current 70 μA) results in T_e of $\sim 1000\text{K}$, as shown in Fig. S8 and discussed in the Supplementary Section 8.]

8. In Fig. 5b, the Off-state mean transistor is off-state, whereas EL emission is on-stage. Due to the off-state and EL emission on-state at the same regime led to a misunderstanding of the paragraph and confusion.

Our response: We thank the Reviewer for the comment. According to this comment to improve the clarity and readability of our manuscript we rewrote the paragraph “Electrically controlled electroluminescence from a sCNT” and the caption for the Figure 5 in the Manuscript.

With the additional experimental work, the significantly expanded supplementary materials and the revised main text we hope to have addressed the concerns of the referee. We hope that she/he will support publication of our revised manuscript in Nature Communications.

Reviewer #2 (Remarks to the Author):

This manuscript from Anna P. Ovvyan et al reports on the electroluminescence of a single semiconducting (9,8) carbon nanotube coupled to a photonic cavity. The main noteworthy result of this manuscript is the deterministic placement of a single (9,8) carbon nanotube inside a photonic crystal cavity using dielectrophoresis.

The electrodes used for the nanotube precise deposition are further used to electrically drive the nanotube luminescence.

The authors chose a cross-bar photonic crystal cavity design. The main advantage with this approach is the direct coupling with photonic waveguide, which allow further integration with photonic circuit. To overcome the optical loss inherent in using metal electrode near the photonic crystal cavity, the authors use graphene electrodes instead, which are demonstrated to be low loss. After a characterization of the cavity using the black-body emission from a graphene strip inserted into the cavity, the authors successfully demonstrate the electroluminescence from the (9,8) nanotube, and its electrical driving.

This is an interesting paper with a strong demonstration of proper electroluminescence from semiconducting carbon nanotube coupled into an easily integrable photonic cavity.

However, a similar work was already demonstrated by the same group (“cavity-enhanced light emission from electrically driven carbon nanotubes” Nature photonics, doi:10.1038/NPHOTON.2016.70), where light emission from carbon nanotube was coupled into a suspended photonic waveguide (nanobeam cavity). The improvement of this work compared to the previous work mainly reside in the clever deposition method of the nanotube, dielectrophoresis using low loss graphene electrode, and the cross-bar cavity design which allow further integration into photonic circuitry.

The use of graphene black body emission by itself was already published in “thermal radiation control from hot graphene electrons coupled to a photonic crystal nanocavity”, Nature communications, doi:10.138/s41467-018-08047-3.

While this paper is properly documented, I don’t feel that there is new insight that justify publication in a broad audience journal such as Nature communications. On the other hand, the technological improvements are interesting and deserve to be published in a more specialized journal.

Should this manuscript be resubmitted elsewhere, I suggest that the authors correct the typo and in particular the errors with the references. In addition, some statistics about the working devices should be provided, such as the ratio of working devices over the fabricated devices, and if possible, the device variability. For example, the device 2 in Fig5 display a significantly lower source-drain current than devices 1 and 3. The authors should comment these points.

Our response: We thank the Reviewer for the comments.

Here, we address the Reviewer’s comments one by one as summarized below.

We would like to note that we performed new experimental measurements to address the concerns of the Reviewer (the experiments required certain amount of time to be accomplished). We added the results in the new Supplementary sections 6-10.

1. The Reviewer mentioned that “The main noteworthy result of this manuscript is the deterministic placement of a single (9,8) carbon nanotube inside a photonic crystal cavity using dielectrophoresis.”

Our response: We politely disagree with the Reviewer. We highlighted and emphasized the novelty of our work below.

First, we demonstrate for the first time coupling of **electroluminescent (EL)** emission of a single sCNT (Fig. 1 d) to the fundamental resonance mode of a PhC cavity (Fig. 5d) in contrast with broadband incandescent emission (thermal radiation) of an electrically-biased metallic CNT, as shown in the work of F. Pyatkov *et al.* [7].

Second, we demonstrate for the first time **full dynamic control with close to 100% on-off ratio (depth) of enhanced EL from a single sCNT (Fig. 1 d) by change of the back-gate voltage** which produces the electric field between the sCNT and the Si substrate (without dissipation of energy). The above-mentioned control is demonstrated in Fig. 5d,e and in the newly added Fig. S6 (new Supplementary Section 6).

Third, we developed a novel **ultra-low-loss electrical contacting scheme based on nanocrystalline graphene electrodes** which provides an elegant way to electrically control photonic devices without inducing absorption or scattering losses (Fig. 2). This allows us to place electrodes directly inside the cavity for the accurate alignment of sCNT in a required region, namely in the maximum of the electric field of the resonance mode (Fig. 4a and Fig. S4a) without degradation of the cavity Q-factor as illustrated in Fig. S2. This leads to the realization of highly efficient EL coupling ($\beta_{int} = 99.5\%$) without compromising the optical quality of the cavity (Fig. S2) with obtained enhancement ($F_{int}=188$) in the telecommunication band (Fig. 5c,f).

Forth, we integrated on chip broadband highly efficient polymer 3D printed couplers (Fig. 1b,c) which are compatible with our hybrid fabrication approach to read out the enhanced EL signal covering the wavelength window from 1400 to 1600 nm. This way the cavity enhanced emission can be tailored to desired spectral regions in devices with a nanoscale footprint.

Thus, we believe that our results and demonstrated full control of a cavity enhanced single electroluminescent emitter will appeal to a broad audience in carbon-based materials, advanced photonics, integrated optics, as well as system integration and engineering. We believe the new approach provides new avenues for nanoscale device design.

In order to emphasize and clarify the novelty of our work, we **modified and added the following paragraphs** in the Abstract, Introduction and Conclusion of the main manuscript text (see below).

[We demonstrate enhanced spectral line shaping of the EL sCNT emission. By back-gating the sCNT-nanoemitter we achieve full on-chip electrical dynamic control of the EL sCNT emission with high on-off ratio and strong enhancement in the telecommunication band. Using nanographene as a low-loss material to electrically contact sCNT emitters directly within a photonic crystal cavity enables realization highly efficient EL coupling without compromising the optical quality of the cavity. Our integrated hybrid approach allows to redirect the enhanced EL signal propagating in photonic waveguide to 3D polymer couplers for convenient out of plane optical read-out. Our versatile procedure of site-selective placement of electroluminescent sCNT emitters in custom-designed low-loss nanographene-photonic environments with high fidelity and reproducibility paves the way for controllable integrated photonic circuits on-chip.]

[We demonstrate enhancement of the EL emission of sCNTs by efficient coupling into a hybrid nanographene-PhC cavity device. Importantly, we obtain full dynamic control of the intensity of the enhanced EL sCNT by active electrical operation of the back-gate voltage, which in fact proves the electroluminescent nature of the emitted sCNT light.]

[The electrically controllable hybrid NCG-Si₃N₄ photonic circuits with deterministic placement of sCNTs in the cavity region provide a scalable and reproducible platform which meets the requirements of integrated photonic circuits for classical and quantum applications.]

[As a proof of principle of our versatile approach of full electrical control of an EL-sCNT, we demonstrate in the Supplementary Section 6 the experimental measurements of the other hybrid device where the enhanced EL-sCNT (which is coupled to a low-loss nanographene-photonic environment) is dynamically operated via back-gate voltage regulation.]

[Notably, the presence of the EL signal at the end of the PhC cavity (red curve in Fig. 5d) and the simultaneous absence of the free-space EL signal detected at the position of the investigated coupled sCNT (green curve in Fig. 5d) in the switched-on state ($U_g=20V$) of the demonstrated device testifies successful coupling of EL into the fundamental resonance mode of the PhC cavity.]

[Cavity enhancement combined with low-loss electrical drive and back-gate configuration electrical control of the nanoemitter sCNTs may allow for realizing tailored single photon sources in a chip-scale framework. With recent advances in boosting the emission rates, our approach to site-selectively integrate sCNTs in telecommunication photonic cavities provides promising avenues for hybrid quantum photonic circuits.]

2. The Reviewer mentioned that “the improvement of this work compared to the previous work from the same group (Fütterling et al “Cavity-enhanced light emission from electrically driven carbon nanotubes” Nature photonics, doi:10.1038/NPHOTON.2016.70) “mainly reside in the clever deposition method of the nanotube, dielectrophoresis using low loss graphene electrode, and the cross-bar cavity design which allow further integration into photonic circuitry.”

Our response: We politely disagree with this point.

Below we highlight the novelty of our work and show the fundamental differences of our work in comparison with mentioned above work of F. Pyatkov *et al.* [Nat. Photon. 10, 420–427 (2016)] [7].

First, F. Pyatkov *et al.* [7] electrically excited **broadband incandescent emission (thermal radiation)** of a biased **metallic CNT**, which was coupled into the nanobeam cavity. In contrast, we integrated **semiconducting CNT** and excited fundamentally different nature of light - **electroluminescence (EL) of sCNT**, and coupled it into Photonic Crystal Cavity. Importantly, we applied three orders of magnitude lower biasing current (nA range) to generate EL sCNT in comparison with incandescent emission in [7].

Second, in contrast with the work [7] we **for the first time experimentally obtained full dynamic control** of the intensity of the **enhanced EL** of a **single sCNT** (Fig. 1d) with 100% on-off ratio by change of back-gate voltage, which produces electric field between sCNT and Si substrate (without dissipation of energy). The above-mentioned control is demonstrated in Fig. 5d,e and in **the newly added Fig. S6. This is direct confirmation of the electroluminescence nature of the coupled sCNT emission.**

Third, we developed a novel, ultra-low-loss electrical contacting scheme based on nanocrystalline graphene electrodes which provides an elegant way to electrically control photonic devices without inducing absorption or scattering losses (as illustrated in Fig. 2) in comparison with metal electrodes utilized

in the work F. Pyatkov *et al.* [7]. This allows to place electrodes directly on the cavity for the accurate alignment of sCNT in a required region, namely in the maximum of electric field of the resonance mode (Fig. 4a and Fig. S4a) without degradation of the cavity Q-factor as illustrated in Fig. S2.

Fourth, importantly, our sCNT-emitter operates in the wide telecommunication window (1400-1550nm) and thus in a technologically viable wavelength range that has not been in reach before, while in the work F. Pyatkov *et al.* [7] working wavelength region is $\lambda=900-1000\text{nm}$.

Fifth, in contrast with work [7] where F. Pyatkov *et al.* utilized fragile and in terms of nanofabrication a freestanding and more complicated (suspended) nanobeam cavity, we developed stable and thus easy to fabricate tailor-made cross-bar PhC cavities on an SiO₂ substrate (Fig. 1d-e and Fig. S1a), leading to experimental high enhancement factor ($F_{int}=188$) and coupling efficiency ($\beta_{int} = 99.5\%$) of EL coupled into resonance modes (Fig 5c,f). This enhancement is more than 30x stronger compared to [7] and substantially more efficient (compared to 30% in [7]).

Sixth, in contrast with work [7], where F. Pyatkov *et al.* utilized standard grating couplers, our developed hybrid photonic devices are equipped with broadband, highly efficient 3D polymer couplers (Fig. 1b) for convenient out of plane optical read out of the enhanced EL signal in a wide telecommunication band. Thus, our hybrid photonic device can be further operated as a broadband emitter covering the wavelength window from 1400 to 1600 nm.

To highlight the differences of our work in comparison with F. Pyatkov *et al.* [Nat. Photon. 10, 420–427 (2016)] [7] we summarized the main above-mentioned points in the table below.

	Our work demonstrated in this Manuscript	F. Pyatkov et al. [Nat. Photon. 10, 420–427 (2016)] [7]
Type of integrated CNT	Semiconducting CNT (sCNT)	(metallic) CNT
CNT detected emission light	Electroluminescent (excitonic) emission (EL)	Incandescent emission
Emission (working) wavelength range	Telecommunication band (1400-1500nm)	Near-visible range 900-1000nm
Control of enhanced CNT emission	First time demonstrated full electrical control of enhanced electroluminescent emission intensity of sCNT by change of back-gate voltage , which produces electric field between sCNT and Si substrate (without dissipation of energy) (field-effect configuration)	Electrical control of incandescent emission via change of bias of CNT. gate was not provided.
Electrical contacting scheme	Novel, ultra-low optical loss electrical contacting scheme based on nanocrystalline graphene	Absorptive metal (gold) electrodes which suppressed resonance modes of the cavity

	electrodes which does not degrade Q-factor of PhC cavity	
Experimentally determined enhancement of the CNT emission	$F_{int}=188$ (Enhanced EL emission)	$F_{on_resonance}=5$ (Enhanced incandescent emission)
Experimentally determined coupling efficiency β_{int} of the CNT emission	$\beta_{int}=99.5\%$ (Coupled EL emission)	$\beta_{int}=30\%$ (Coupled incandescent emission)
Type of the cavity	Not-freestanding and easy in fabrication cross-bar PhC cavity on SiO ₂ substrate	Fragile Freestanding (suspended) nanobeam PhC cavity
Couplers for out of plane read-out optical enhanced emission signal	High-efficiency broadband 3D polymer couplers covering wide telecommunication range [18]	Grating couplers

In order to highlight the fundamental difference of our work in contrast with F. Pyatkov *et al.* [Nat. Photon. 10, 420–427 (2016)] [7] we added the following sentences in the main text of the manuscript (see below).

[The applied low driving bias-current (tens of nanoamperes) to the sCNT further proves the electroluminescent nature of the emitted light, in contrast to previous work by Pyatkov et al. [7] where three orders of magnitude higher biasing current was utilized to generate incandescent emission of CNTs. We achieve an on-off ratio close to 100%.]

[Importantly, the utilized biasing current of the sCNT is three-four orders of magnitude lower in comparison with the nanocrystalline graphene strip thermal emitter described in the previous paragraph, as well as in comparison with the incandescent CNT realized in the work of Pyatkov et al. [7]. This again confirms the EL nature of our sCNT emission in contrast with incandescent NCG emission..]

3. The Reviewer mentioned: “The use of graphene black body emission by itself was already published in “thermal radiation control from hot graphene electrons coupled to a photonic crystal nanocavity”, Nature communications, doi:10.138/s41467-018-08047-3.”

Our response: We thank the Reviewer for the comment.

We highlight below the differences of our work in comparison with Shiue *et al.* [Nat. Commun. 10, 109 (2019)] [21].

We incorporated a thermal nanocrystalline graphene strip nanoemitter in a predetermined position within a photonic device which allows us to probe the LDOS factor and provides optimal coupling of emitted light into cavity modes at cryogenic and room temperature. We would like to highlight that in contrast with the work of Shiue *et al.* [21], we demonstrated a strip-nanoemitter which can be integrated on nanophotonic devices **for high-resolution LDOS sensing owing to nano-dimensions and importantly**

without degradation of quality factor of the cavity (Fig. 2, Fig. 3 and Fig S2, S3) at cryogenic and room temperature. Furthermore, the emitter can be easily removed by oxygen plasma (while protecting the rest of the circuit with photoresist, for example). We also note, that in the work of Shiue *et al.* [21] ~ three-four-times degradation (drop) of Q-factor of 2D PhC cavity was found after graphene deposition on top as well as red-shift of the resonance modes. In our case, our approach maintains high quality factors due to negligible influence of the NCG electrodes on the cavity mode.

To highlight the difference of our work in comparison with Shiue *et al.* [21] we added the following sentences in the main text of the manuscript (see below).

[Such a thermal strip-nanoemitter incorporated in a predetermined position within a photonic device allows to probe the LDOS factor and provides optimal coupling of emitted light in cavity modes at cryogenic and room temperature. The nanocrystalline graphene strip nanoemitter can be removed by oxygen plasma (while protecting the rest of the circuit with photoresist, for example) and thus makes it a promising candidate for optical high-resolution LDOS sensing at room and cryogenic temperature. Importantly, NCG strip-nanoemitter doesn't degrade quality factor of the cavity as shown in Fig. S2c in contrast with the work of Shiue *et al.* [21], where it is demonstrated a micro black-body radiator for optical communication application.]

4. The Reviewer has mentioned "I suggest that the authors correct the typo and in particular the errors with the references".

Our response: We thank the Reviewer for the suggestion. We have corrected the typo and the Reference list.

5. The Reviewer has mentioned "In addition, some statistics about the working devices should be provided, such as the ratio of working devices over the fabricated devices, and if possible, the device variability. For example, the device 2 in Fig5 display a significantly lower source-drain current than devices 1 and 3. The authors should comment these points."

Our response: We thank the Reviewer for the comment. We address it below.

In our experiments, we detected successful coupling of sCNT EL emission into resonance modes in all hybrid NCG-PhC cavity devices which we have measured. The tailor-made cross-bar PhC cavities on the chip contain various periods ($a=453\text{nm}-466\text{nm}$) and number of hole segments ($N=25-45$), which ensured variance of the resonance modes in the spectral range between 1400nm-1500nm to match (9,8)-sCNT (central wavelength 1440 nm) emission line in the telecommunication band [8]. Our effective fabrication yield of NCG-Si₃N₄ devices is 95%. Electric field-assisted dielectrophoresis provides a stable deposition method for sCNTs with an effective yield of 83% in our case. Our optimized full fabrication protocol enables reproducibility for photonic applications in the telecommunication band. We added the new **Supplementary section 10** with the photograph and micrograph of the fabricated chip.

We address the comment of the Reviewer regarding *the device 2 in Fig. 5* below.

We reported in the manuscript (Fig. 5c-f and in the newly added Fig. S6) electrically-driven sCNTs which emit EL emission coupled to PhC cavity devices on-chip, where we realized full dynamic electrical control of the EL intensity via back gate-voltage to prove the electroluminescent nature of the emitted light. Notably, with increase of the source-drain current (higher than several hundreds of nanoampere) the

probability of thermal emission of sCNT rises. In case of the *device 2* (Fig. 5c,f), thermal emission of the integrated sCNT starts to appear at an I_{sd} around 100nA, which is evident due to the widening in its emission spectrum, in contrast with *devices 1,3*. Therefore, for the *device 2* we included β_{int} and F_{int} values only for the biasing $I_{sd} < 100\text{nA}$ when only EL of the sCNT was excited in the device. In contrast, for the *devices 1* and *3*, thermal emission starts to appear at biasing currents higher than several hundreds of nanoampere. Thus, β_{int} and F_{int} in Fig. 5c,f demonstrate successful coupling of EL emission into a tailor-made PhC cavity.

In order to prove the reproducibility of our approach, we **performed anew measurements** and demonstrated again full electrical control of the intensity of enhanced electroluminescence from a single sCNT coupled to a hybrid nanographene-PhC cavity device. We summarized the results in the **new Supplementary Section 6**. This confirms the fact that our novel method is scalable to arrays of devices, taking into account also the measurements which are shown in Fig. 5 c-f.

We added new the Supplementary Section 6 (see below):

[**6. Dynamic control of EL from cavity-integrated sCNT**

As a proof of principle of our versatile approach of full electrical control of single sCNT nanoemitter integrated in custom-designed low-loss nanographene-photonic environment, we demonstrate in this section the measurements of the other hybrid device, where sCNT EL coupled to nanographene-PhC cavity is dynamically operated by regulation of back-gate voltage.

Electrically-biased sCNT integrated in the cross-bar PhC cavity with simultaneously applied back-gate voltage ($U_g = -25\text{V}$) leads to an enhanced excitonic EL emission coupled into the odd resonance modes of the cavity in agreement with LDOS-spatial maps (Fig. 4a, S4a). The measured spectrum of light outcoupled from one of the ends (coupler C in Fig. 1c) of the investigated hybrid device is shown in Fig. S6a (red curve). This is obtained when sCNT is charged neutral, which corresponds to the EL *switched-on* state. Changing the gate voltage from -25V to +30V leads to decrease of intensity of the excitonic EL emission which is further coupled to resonance modes and outcoupled via coupler C (light and dark green curves in Fig. S6a). Switching the gate voltage to $U_g = +30\text{V}$ leads to complete suppression of the excitonic EL emission, corresponding to EL *switched-off* state (blue curve in Fig. S6a). Thus, we obtain dynamic control of sCNT enhanced EL emission with close to 100% on-off ratio (depth) via active electrical operation of back-gate voltage.

Figure S6. Experimental dynamic control of EL from cavity-integrated sCNT. *a) The spectra of EL sCNT coupled to odd resonance modes at 1419.5nm, 1434.4nm, 1452.9nm, acquired from coupler C of NCG-Si₃N₄ PhC cross-bar device gated with corresponding voltage U_g . At $U_g = -25V$ – EL is in switched-on state (red curve) and at $U_g = +30V$ – EL is switched-off (blue curve), sCNT biasing current is constant $I_{sd} = 30nA$. Cross-bar PhC cavity consists of $N = 45$ segments in each Bragg mirror with a lattice period of $a = 457nm$. The resonance modes are labelled. **b) U_{sd} - U_g curve acquired at constant sCNT biasing current $I_{sd} = 30nA$. Forward and backward sweep traces are shown. The transport data shows negligible hysteresis between the forward and backward sweeps due to the cryogenic environment (77 K). The yellow area corresponds to the regime, in which the excitonic EL emission is in the switched-on state. All data recorded at 77K.]***

To refer to the Supplementary Section 6, we added the following sentence in the main text of the manuscript:

[As a proof of principle of our versatile approach of full electrical control of an EL-sCNT, we demonstrate in the Supplementary Section 6 the experimental measurements of the other hybrid device where the enhanced EL-sCNT (which is coupled to a low-loss nanographene-photonic environment) is dynamically operated via back-gate voltage regulation.]

Furthermore, we added the new **Supplementary Section 10** with the photograph and micrograph of the fabricated chip (see below).

[10. Hybrid NCG-Si₃N₄ PhC devices on-chip

In the experiment we detected successful coupling of sCNT EL emission into resonance modes in all hybrid NCG-PhC cavity devices which we have measured. The tailor-made cross-bar PhC cavities on the chip consist various periods ($a = 453nm - 466nm$) and number of the segments ($N = 25 - 45$), which ensured variance of the resonance modes in wide range 1400nm-1500nm to match (9,8)-sCNT (central wavelength 1440 nm) emission line in the telecommunication band [6]. Our effective fabrication yield of NCG-Si₃N₄ devices is close to 95%. Electric field-assisted dielectrophoresis provides a stable deposition method for sCNTs with the effective yield close to 83% in our case. Our optimized full fabrication protocol enables reproducibility for photonic applications in the telecommunication band.

Figure S10. Fabricated hybrid devices on-chip. *a) Photograph of the chip (1cm^2) for full dynamic electrical control of enhanced sCNT electroluminescent emission via operation of back-gate voltage. The pattern dimension is 5.4mm^2 . b) Optical micrograph of NCG-Si₃N₄ PhC cavity devices equipped with 3D couplers. Scale-bar: $200\ \mu\text{m}$.]*

In order to demonstrate deterministic placement of individual sCNT provided by dielectrophoresis, we included a new **SEM image** of the investigated single sCNT deposited between NCG electrodes in the cavity region (by site-selective dielectrophoresis) of the experimentally measured hybrid device. **We added this SEM image in Fig. 1d with the inset showing a close-up image of an individual sCNT in the cavity.** We dynamically control EL of this individual sCNT which is demonstrated in Fig. 5b,d,e.

Modified Figure 1 in the Manuscript with the new SEM image of single sCNT in the cavity region in (d) is shown below.

Figure 1. Electrically controlled cross-bar PhC cavity with integrated sCNT emitter. **a)** Schematic of the hybrid device. The sCNT is positioned between nanocrystalline graphene electrodes placed on top of the cavity crossing. The nanographene electrodes are used for addressing the sCNT and are connected to metallic leads for current injection. The electroluminescent emission is coupled out from both ends of the PhC cavity. **b)** Helium Ion microscopy image of a broadband total reflection 3D coupler connected to the ends of the nanophotonic waveguides. **c)** Optical micrograph of the fabricated device with 3D output couplers and cavity region marked by the zoom-box. **d)** Scanning electron microscope image of PhC cavity with single sCNT integrated between nanocrystalline graphene electrodes. The inset shows a close-up image of an individual sCNT in the cavity. **e)** Scanning electron microscope image of the hybrid device in the cavity region.]

We also added the following sentence in the Manuscript text:

[Controllable integration of single sCNT in the required location is achieved by self-alignment between electrodes through the induced electrostatic field during dielectrophoresis. Thus, owing to the design, a single sCNT is placed in the optimal position in the center of the cavity as shown in Fig. 1d (see inset).]

Thus, dielectrophoresis provides a stable and deterministic deposition method for single sCNTs between NCG electrodes on hybrid devices. Successful deposition is obtained with optimized parameters of the applied electric field as well as an optimized concentration of sCNT in diluted suspension. Furthermore, to confirm the deposition of individual sCNTs, transport characteristics of the devices were carried out at ambient conditions in a probe station with TRIAX probes using an Agilent 4155C semiconductor parameter analyzer. The detailed recipe of sCNT deposition and parameters of Dielectrophoresis are reported in the Methods and Fabrication section.

With the additional experimental work, the significantly expanded supplementary materials and the revised main text we hope to have addressed the concerns of the referee. We hope that she/he will support publication of our revised manuscript in Nature Communications.

Reviewer #3 (Remarks to the Author):

Anna P. Ovyvan et al report a paper on “A cavity enhanced and tunable electroluminescent carbon nanotube emitter in the telecom band”. The authors propose an original device configuration where a single CNT is coupled to nanographene electrodes for the carrier injection and to a photonic cavity for the extraction of photons. This topic is very interesting and the results are convincing. Nevertheless, I have few concerns before I can recommend this paper for publication in Nature Communications:

Our response: We thank the Reviewer for the positive comments on our novel work.

Here, we address the Reviewer’s comments and questions one by one summarized below.

We performed new experimental measurements to address the concerns of the Reviewer (the experiments required certain amount of time to be accomplished), we added the results in the new Supplementary sections 6-10.

1- The authors mention several times in the main text the Purcell effect. Nevertheless, there is no experimental proof that the Purcell regime is achieved here. This takes nothing away from the interest of the results reported in the paper. But, the author should stay with enhancement factors, meaning enhanced collection of photons due to the redirection in the optical mode of the cavity.

Our response: We thank the Reviewer for the suggestion. According to the suggestion of the Reviewer to be rigorous we stay with “*enhancement factor*”, and we made changes in the main Manuscript text (see below).

[We demonstrate enhanced spectral line shaping of the EL sCNT emission. By back-gating the sCNT-nanoemitter we achieve full on-chip electrical dynamic control of EL sCNT emission with high on-off ratio and strong enhancement in the telecommunication band.]

[We demonstrate enhancement of the EL emission of sCNTs by efficient coupling into a hybrid nanographene-PhC cavity device.]

[We achieve an *on-off* ratio close to 100%. In the *switched-on* regime, we find a high enhancement factor up to $F_{int}=188$ and coupling efficiency $\beta_{int}=99.5\%$ of the EL into the fundamental resonance mode, which is efficiently read out by 3D couplers terminating the PhC cavity.]

[In addition, we show that the nanocrystalline graphene strip in the cavity region can function as a thermal nanoemitter, thus providing telecom-band polarized emission with a peak enhancement factor $F=80$ and $F=112$ at 77K and 300K, respectively.]

[The cross-bar PhC cavity provides on-resonance enhancement of NCG incandescent emission evanescently coupled into odd modes, obtaining maximum values $F_{III}=80.6$ at cryogenic and $F_{III}=112.7$ and room temperature.]

[The full-bandwidth enhancement-factor F_{int} and coupling efficiency β_{int} of the emitted EL signal are experimentally determined in the same fashion as with the incandescent NCG strip, integrated over the full wavelength of the signal.]

We would like also to comment on the Purcell and LDOS enhancement factors below.

According to the Purcell theory [15], enhanced light-matter interaction with the emitter is achieved when placing the source in the antinode of the electric field, where the emitter's wavelength and polarization need to match the resonance mode of the PhC cavity.

Owing to our novel cross-bar PhC cavity design, we satisfied the above mentioned requirements, namely a single sCNT is placed in the optimal position in the center of the cavity (Fig. 1d), where the theoretical maximum enhancement of EL coupled to the tailored fundamental resonance mode is expected, according to the simulated LDOS enhancement spatial maps shown in Fig. 4a and Fig. S4a. The Purcell enhancement factor approximates the single-resonant LDOS enhancement. This shows that the EL of sCNT experienced Purcell enhancement in our experiments (Fig. 5d, Fig. S5b and new added Fig. S6). Similarly, we also obtained Purcell enhancement of the incandescent emission of the NCG- strip thermal source integrated in the center of the cavity (Fig. 3c) coupled to odd resonance modes of the PhC cavity (Fig. 3d, Fig. S3b).

Notably, in the experiment we determined the full-bandwidth enhancement-factor F_{int} and coupling efficiency β_{int} of the emitted EL sCNT signal, integrated over the full wavelength of the signal. We demonstrated enhanced spectral line shaping of the EL sCNT emission with a high enhancement factor up to $F_{int}=188$ and coupling efficiency $\beta_{int}=99.5\%$ of the EL into the fundamental resonance mode (Fig. 5d,c,f).

2- On figure 5d), the EL spectra in different experimental conditions are plotted. In particular, if I understand, the green curve is supposed to show the one of the CNT detected in the confocal configuration (so not coupled to the cavity). Nevertheless, nothing is visible on the figure. So, is there any EL signal of the CNT?

Our response: We thank the Reviewer for this comment and we clarify this point below.

We **simultaneously** detected enhanced EL signal from coupler C (red curve in Fig. 5d) and free-space EL emission at the position of the **coupled sCNT** (green curve in Fig. 5d) on the same hybrid device (shown in Fig. 1d) **at the same experimental conditions at $U_g=20V$.**

Absence of the signal detected at the coupled sCNT position and simultaneous presence of the EL signal at the end of PhC cavity of the demonstrated device proves successful coupling of EL into the fundamental resonance mode of the PhC cavity.

In order to clarify this point, we added the following sentence in the Manuscript (see below).

[Notably, the presence of the EL signal at the end of the PhC cavity (red curve in Fig. 5d) and the simultaneous absence of the free-space EL signal detected at the position of the investigated coupled sCNT (green curve in Fig. 5d) in the switched-on state ($U_g=20V$) of the demonstrated device testifies successful coupling of EL into the fundamental resonance mode of the PhC cavity.]

We also edited the caption in Fig. 5d (see below).

[

Figure 5. Dynamic control of EL from cavity-integrated sCNT. **a)** Simulated spectrum of light coupled out from one of the ends the cross-bar PhC cavity. The symmetric design of the PhC cavity leads to equal EL emission at both sides. **b)** U_{sd} - U_g curve acquired at constant sCNT biasing current $I_{sd} = -90$ nA. Forward and backward sweep traces are shown. The yellow area corresponds to the regime, in which the excitonic EL emission is in the *switched-on* state. The red point corresponds to the electric acquisition point of the spectrum (red curve) shown in d). **c)** Measured dependence of β_{int} on I_{sd} biased sCNT for several PhC devices. **d)** The spectrum of EL sCNT coupled to the fundamental resonance mode at 1438.1 nm, acquired from coupler C of a PhC cross-bar device gated with $U_g = 20$ V (EL *switched-on* state, red curve) and at $U_g = 0$ V (EL *switched-off* state, blue curve). The spectrum of free-space EL simultaneously recorded at the position of the sCNT coupled to the cavity in the *switched-on* state at $U_g = 20$ V of the same investigated device is shown by green curve. sCNT biasing current $I_{sd} = -90$ nA. **e)** EL excitation map of enhanced EL intensity integrated over the fundamental resonance mode (1430-1450 nm) outcoupled from coupler C of explored device as a function of driven I_{sd} and U_g . **f)** Measured dependence of F_{int} on I_{sd} applied to the sCNT for three cross-bar PhC devices, acquired at 77K. Device 1 consists of $N=25$ segments in each Bragg mirror with a lattice period of $a=462$ nm; device 2: $N=35$, $a=460$ nm; device 3: $N=35$, $a=455$ nm. All data recorded at 77K.]

3- The authors, show also EL from the nanographene electrodes. If I am right, they observe it at much higher current than for the EL of the CNT. Is it right? If yes, I suggest that they insist on this point in the text.

Our response: We thank the Reviewer for the questions.

Nanocrystalline graphene (NCG) is a *zero-bandgap* material of semi-metallic nature, therefore it cannot emit electroluminescence or photoluminescence. Instead, **NCG is a thermal emitter and thus emits incandescent light** with blackbody characteristics through Joule heating under electrical excitation. (We determined the electron temperature of electrically-biased NCG and added the results in the new Supplementary Section 8.)

In contrast, a semiconducting CNT (sCNT) is a 1D nanomaterial and can be modelled as a graphene layer rolled-up into a cylinder, which introduces/imposes a periodic boundary condition, resulting in the opening of a direct bandgap (it scales roughly as the inverse of its diameter with typical values of 1 eV for $d \approx 1$

nm), according to the zone-folding theory. Therefore, a **sCNT can emit electroluminescence** and photoluminescence via carrier recombination under electrical and optical excitation, respectively.

Regarding the biasing current (source-drain current) of sCNT and NCG strip, we would like to point out the following. Three-four orders of magnitude higher applied source-drain current is required to excite incandescent emission of the NCG strip through Joule heating in comparison with the excitation of electroluminescence of the sCNT reported in the manuscript, due to the different nature of both emissions. In the case of the NCG strip, the supplied electrical energy is transformed into Joule heat and dissipated in the NCG leading to incandescent emission. In contrast, electroluminescent photons in electrically-driven sCNTs are generated through radiative recombination of excitons formed by injected carriers (electrons and holes) within the nanotube under three-four orders of magnitude lower applied source-drain current (10-160nA).

Further evidence of the electroluminescent nature of the sCNT emission is the demonstrated full dynamic control of the EL emission intensity of the electrically-biased sCNT by applying a gate voltage between the sCNT and the Si substrate in the back-field configuration (Fig. 1a). In contrast, incandescent emission cannot be controlled via a back-gate voltage, which is also apparent from the independence of the source-drain current of the NCG nano-strip from the back-gate voltage, shown in Fig. S9 in the **newly added Supplementary Section 9**, in comparison with the transport curve of the sCNT (Fig. 5b, Fig. S6b).

According to the Reviewer's suggestion to highlight this point we added the following sentences in the Manuscript (see below).

[We emphasize that in the case of the demonstrated biased NCG nano-strip emitter, the supplied electric energy is transformed into Joule heat and is dissipated in the nanocrystalline graphene leading to incandescent emission. Notably, the determined electron temperature of the NCG-based strip-emitter on Si₃N₄ waveguide at an applied electrical power of 2.87 mW (source-drain current 70 uA) results in T_e of $\sim 1000\text{K}$, as shown in Fig. S8 and discussed in the Supplementary Section 8. Thus, we note that the electrical excitation of the thermal strip-nanoemitter (hundreds of microamperes) is three-four orders of magnitude higher in comparison to the electroluminescent sCNT reported in the next paragraph, due to the incandescent nature of the NCG emission. Thermal radiation of the electrically-biased NCG nano-strip source is confirmed by the measured independence of the source-drain current through the NCG during the change of the back-gate voltage, shown in Fig. S9.]

[Importantly, the utilized biasing current of the sCNT is three-four orders of magnitude lower in comparison with the nanocrystalline graphene strip thermal emitter described in the previous paragraph, as well as in comparison with the incandescent CNT realized in the work of Pyatkov et al. [7]. This again confirms the EL nature of our sCNT emission in contrast with incandescent NCG emission.]

We added the new Supplementary Section 9 (see below):

[9. Independence of source-drain current on NCG-based incandescent nanoemitter from gate voltage

Hybrid device with NCG strip incorporated in the cavity region between NCG electrodes (Fig. 3a,c) is a thermal nano-source which allows us to probe the LDOS factor and provides optimal coupling of emitted light in cavity modes at cryogenic and room temperature, it is investigated and demonstrated in Fig. 3 and Fig.S3. We emphasize that supplied electric energy to NCG strip is transformed into Joule heat and dissipated in NCG, which emits incandescent emission. Thus, we note that the electrical excitation of the

thermal NCG strip nanoemitter (biasing current ~ hundreds of microamperes) is three-four orders of magnitude higher in comparison to the electroluminescent sCNT due to the incandescent nature of the NCG emission. To prove incandescent nature of the emitted light from biased NCG nano-strip we demonstrate independence of source-drain current on NCG on the change of gate voltage, which is shown in Fig. S9.

Figure S9. Measured independence of NCG strip source-drain current on gate voltage. The NCG biasing voltage is 37V (a) and 5V(b).]

We added the new **Supplementary Section 8** (see below):

[8. Telecom NCG-based incandescent nanoemitter electron temperature

We experimentally measured free-space incandescent emission spectrum of electrically-biased nanocrystalline graphene strip placed on Si₃N₄ waveguide (similar device design as shown in Fig. 3c, but NCG strip is placed on waveguide instead of cavity). The detected monotonic thermal spectrum (red curve) is shown in Fig. S8.

Importantly, in our experiments (Fig. 3 and Fig. S8), utilized NCG strip (thickness 5 nm) consists ~ 15 layers of single-layer graphene, and it is placed on Si₃N₄ cavity or waveguide (NCG strip is not suspended). Both of these facts allow us to assume that NCG strip infrared emission represents an electronic temperature, since there are no significant non-equilibrium phonon distribution exists [3] [4].

Thus, we fit incandescent emission of biased NCG strip at applied electrical power 2.87 mW (red curve in Fig. S8a) with a grey-body theory (Planck’s law, modified by an emissivity) (1), and determine electron temperature of NCG – T_e of ~ 1000K, as indicated by dashed blue curve in Fig. S8.

$$I(\lambda, T) = \varepsilon * \frac{2\pi hc^2}{\lambda^5} * \frac{1}{\exp\left(\frac{hc}{\lambda k_B T} - 1\right)}, \quad (1)$$

where $I(\lambda, T)$ – spectral energy density of thermal radiation from NCG strip, h – Planck constant $6.626 * 10^{-34}$ (J * sec), k_B – Boltzmann const $1.380 * 10^{-23}$ ($\frac{J}{K}$), T – electron temperature (K) of NCG strip, λ – wavelength of the emitted photons, c – speed of light, ε – emissivity of NCG strip ($\varepsilon \approx 0.25$) [4] [5].

Figure S8. Emission spectrum of NCG strip nano-emitter on Si_3N_4 waveguide. a) Recorded free-space thermal emission spectrum of electrically-biased NCG strip using 70 μA (source-drain voltage 41V). The spectrum from the NCG strip was acquired with the polarizer parallel to TE mode of the waveguide. The waist of NCG strip is 250 nm in width, thickness – 5 nm. Drop of the detected intensity at $\lambda=1600\text{nm}$ is due to cut-off of sensitivity of utilized InGaAs photodiode linear array. Blue dashed curve: grey-body fit spectrum at extracted temperature $T=1000\text{K}$ according to equation (1), where full fit spectrum is shown in **b**).]

To refer to the Supplementary Section 8, we added the following sentence in the main text of the manuscript:

[Notably, determined electron temperature of NCG-based strip-emitter on Si_3N_4 waveguide at applied electrical power 2.8 mW (source-drain current 70 μA) results in T_e of $\sim 1000\text{K}$, as shown in Fig. S8 and discussed in Supplementary Section 8.]

4- In a previous work, the authors demonstrated the single photon emission of CNT with an electrical injection. Did the authors tried to perform correlation measurements in their devices? This would greatly enhance the interest of the paper.

Our response: We thank the Reviewer for the comment. Goal of our work was the achievement of full control of the intensity of a cavity enhanced electroluminescence sCNT nanoemitter with *on-off* extinction ratio close to 100%. To the best of our knowledge, we firstly demonstrated full turn on-off of the emission of a hybrid sCNT-NCG PhC cavity device, which was not demonstrated in the work of S. Khasminskaya *et al.* [4].

Furthermore, we performed experiments at higher temperatures (77K and 300K) in comparison with the work of S. Khasminskaya *et al.* [4] (2K). To achieve reliable operation of single-photon *detectors* (SNSPDs) the base environment temperature should be below 2K. Due to the higher operating temperature, our devices on-chip were not equipped with SNSPDs, which are necessary to perform autocorrelation measurements in a Hanbury–Brown–Twiss configuration of sCNT emitter with a very short internal exciton lifetime (tens of picosecond), which is reduced even further by placing sCNT in PhC cavity due to EL enhancement.

5- Finally, the authors mention the possibility of scalability of their technics. Here, “only” 3 devices are shown. I would suggest that the authors discuss the scalability at this step of their project. How is efficient the placement of the CNT? When a CNT is well placed on the cavity, how many have the good exciton energy to be in resonance? Etc...

Our response: We thank the Reviewer for the comment. Electric field-assisted dielectrophoresis provides a stable deposition method for sCNTs and allows deterministic placement between electrodes, thus individual sCNTs are bridged between multiple nanocrystalline graphene contact pairs with **effective yield close to 83% in our case**. The successful deposition of single sCNTs between electrodes is accomplished with an optimized applied electric field as well as an optimized concentration of sCNT in diluted suspension (these points are discussed in the Methods section). The yield of our effective dielectrophoresis deposition is in agreement with the demonstrated high yield of bridging CNTs between electrodes in the previous works Ganzhorn *et al.* (Adv.Mater. 2011, 23, 1734–1738) [90 out of 100 devices] and Vijayaraghavan *et al.* (ACS Nano 2010, 4, 5, 2748–2754) [85%].

In the experiment we detected successful coupling of sCNT EL emission into the resonance modes of all measured hybrid NCG-PhC cavity devices. The tailor-made cross-bar PhC cavities on the chip span various periods ($a=453\text{nm}-466\text{nm}$) and number of the hole segments ($N=25-45$), which ensured variance of the resonance modes in wide spectral range of $1400\text{nm}-1500\text{nm}$ to match the (9,8)-sCNT (central wavelength 1440 nm) emission line in the telecommunication band. Our effective fabrication yield of NCG-Si₃N₄ devices is close to 95%. Thus, our optimized full fabrication protocol enables reproducibility and potential scalability for photonic applications in the telecommunication band.

In order to prove reproducibility and potential scalability, we performed new measurements and demonstrated again full electrical control of intensity of enhanced electroluminescence single sCNT coupled to hybrid nanographene-PhC cavity device. The dynamic control with close to 100% on-off ratio (depth) was obtained by change of the back-gate voltage which produces an electric field between the sCNT and the Si substrate. We summarized the results in the **new Supplementary Section 6**. This confirms the fact that our novel method is scalable on the arrays of the devices taking into account also the measurements which are shown in Fig. 5 c-f.

We added new the Supplementary Section 6 (see below):

[6. Dynamic control of EL from cavity-integrated sCNT

As a proof of principle of our versatile approach of full electrical control of single sCNT nanoemitter integrated in custom-designed low-loss nanographene-photonic environment, we demonstrate in this section the measurements of the other hybrid device, where sCNT EL coupled to nanographene-PhC cavity is dynamically operated by regulation of back-gate voltage.

Electrically-biased sCNT integrated in the cross-bar PhC cavity with simultaneously applied back-gate voltage ($U_g = -25\text{V}$) leads to an enhanced excitonic EL emission coupled into the odd resonance modes of the cavity in agreement with LDOS-spatial maps (Fig. 4a, S4a). The measured spectrum of light outcoupled from one of the ends (coupler C in Fig. 1c) of the investigated hybrid device is shown in Fig. S6a (red curve). This is obtained when sCNT is charged neutral, which corresponds to the EL *switched-on* state. Changing the gate voltage from -25V to $+30\text{V}$ leads to decrease of intensity of the excitonic EL emission which is further coupled to resonance modes and outcoupled via coupler C (light and dark green curves in Fig. S6a). Switching the gate voltage to $U_g = +30\text{V}$ leads to complete suppression of the excitonic EL emission, corresponding to EL *switched-off* state (blue curve in Fig. S6a). Thus, we obtain dynamic control of sCNT enhanced EL emission with close to 100% on-off ratio (depth) via active electrical operation of back-gate voltage.

Figure S6. Experimental dynamic control of EL from cavity-integrated sCNT. a) The spectra of EL sCNT coupled to odd resonance modes at 1419.5nm, 1434.4nm, 1452.9nm, acquired from coupler C of NCG-Si₃N₄ PhC cross-bar device gated with corresponding voltage U_g . At $U_g = -25V$ – EL is in switched-on state (red curve) and at $U_g = +30V$ – EL is switched-off (blue curve), sCNT biasing current is constant $I_{sd} = 30nA$. Cross-bar PhC cavity consists of $N = 45$ segments in each Bragg mirror with a lattice period of $a = 457nm$. The resonance modes are labelled. **b)** U_{sd} - U_g curve acquired at constant sCNT biasing current $I_{sd} = 30nA$. Forward and backward sweep traces are shown. The transport data shows negligible hysteresis between the forward and backward sweeps due to the cryogenic environment (77 K). The yellow area corresponds to the regime, in which the excitonic EL emission is in the switched-on state. All data recorded at 77K.]

To refer to the Supplementary Section 6, we added the following sentence in the main text of the manuscript:

[As a proof of principle of our versatile approach of full electrical control of an EL-sCNT, we demonstrate in the Supplementary Section 6 the experimental measurements of the other hybrid device where the enhanced EL-sCNT (which is coupled to a low-loss nanographene-photonic environment) is dynamically operated via back-gate voltage regulation.]

We would like to highlight that in order to prove the deposition method of sCNT in the optimal position on the cavity region leading to high enhancement factor of EL, we took a new SEM image of the investigated single sCNT deposited between NCG electrodes in the cavity region (by site-selective dielectrophoresis) of the experimentally measured hybrid device. We added this SEM image in Fig. 1d with the inset showing a close-up image of an individual sCNT in the cavity. We dynamically control EL of this individual sCNT which is demonstrated in Fig. 5b,d,e.

The modified Figure 1 in the Manuscript with the new SEM image of single sCNT in the cavity region in (d) is shown below.

[

Figure 1. Electrically controlled cross-bar PhC cavity with integrated sCNT emitter. **a)** Schematic of the hybrid device. The sCNT is positioned between nanocrystalline graphene electrodes placed on top of the cavity crossing. The nanographene electrodes are used for addressing the sCNT and are connected to metallic leads for current injection. The electroluminescent emission is coupled out from both ends of the PhC cavity. **b)** Helium Ion microscopy image of a broadband total reflection 3D coupler connected to the ends of the nanophotonic waveguides. **c)** Optical micrograph of the fabricated device with 3D output couplers and cavity region marked by the zoom-box. **d)** Scanning electron microscope image of PhC cavity with single sCNT integrated between nanocrystalline graphene electrodes. The inset shows a close-up image of an individual sCNT in the cavity. **e)** Scanning electron microscope image of the hybrid device in the cavity region.]

We also added the following sentence in the main text:

[Controllable integration of a single sCNT in the required location is achieved by self-alignment between electrodes through the induced electrostatic field during dielectrophoresis. Thus, owing to the design, a single sCNT is placed in the optimal position in the center of the cavity as shown in Fig. 1d.]

Furthermore, we added the new **Supplementary Section 10** with the photograph and micrograph of the fabricated chip (see below).

[10. Hybrid NCG-Si₃N₄ PhC devices on-chip

In the experiment we detected successful coupling of sCNT EL emission into resonance modes in all hybrid NCG-PhC cavity devices which we have measured. The tailor-made cross-bar PhC cavities on the chip consist various periods ($a=453\text{nm}-466\text{nm}$) and number of the segments ($N=25-45$), which ensured variance of the resonance modes in wide range $1400\text{nm}-1500\text{nm}$ to match (9,8)-sCNT (central wavelength 1440 nm) emission line in the telecommunication band [6]. Our effective fabrication yield of NCG-Si₃N₄ devices is close to 95%. Electric field-assisted dielectrophoresis provides a stable deposition method for sCNTs with the effective yield close to 83% in our case. Our optimized full fabrication protocol enables reproducibility for photonic applications in the telecommunication band.

Figure S10. Fabricated hybrid devices on-chip. *a) Photograph of the chip (1cm^2) for full dynamic electrical control of enhanced sCNT electroluminescent emission via operation of back-gate voltage. The pattern dimension is 5.4mm^2 . b) Optical micrograph of NCG-Si₃N₄ PhC cavity devices equipped with 3D couplers. Scale-bar: $200\ \mu\text{m}$.]*

We believe that achieving full control of the cavity-enhanced single sCNT EL performed via change of the back-gate voltage with high on-off ratio in the telecommunication window via a designed ultra-low-loss electrical contacting scheme based on NCG electrodes ensures an elegant way to electrically control photonic devices without inducing absorption or scattering losses and provides promising avenues for hybrid quantum photonic circuits.

With the additional experimental work, the significantly expanded supplementary materials and the revised main text we hope to have addressed the concerns of the referee. We hope that she/he will support publication of our revised manuscript in Nature Communications.

REVIEWERS' COMMENTS

Reviewer #1 (Remarks to the Author):

In the revised manuscript. The authors executed more experiments and support their claims.

The authors addressed my concerns properly. Thus I recommend the publication to Nature Communications.

Reviewer #2 (Remarks to the Author):

I have reviewed the revised version of the manuscript from P. Ovvyan et al. entitled "An electroluminescent and tunable cavity-enhanced carbon-nanotube-emitter in the telecom band". I want to acknowledge the work done behind the new experimental measurements, and I thank the author for this new version, as well as the very clear and precise reply to comments.

I appreciate the additional comment on the statistics, and the new supplementary section 10, as well as the new supplementary section 6. It is now very clear for me that this is a very good manuscript from a technical point of view that deserves to be published.

The question remains if this manuscript will appeal to a broad audience journal such as Nature Communications, and this is really an editorial line question. From my point of view as a referee, I feel now that this is the case. In their reply, the authors made a strong claim to emphasize the novelty of their work and the difference with their previous works, and underlined several points to demonstrate the novelty of this work.

Even if each point taken separately is not a breakthrough, all these points taken together surely justify sufficient new insight to deserve publication in Nature Communications, and I think that this work will appeal to a broader community.

Reviewer #3 (Remarks to the Author):

The referee thanks the authors for their detailed answers and for the addition of the new data and more precise explanations. I have also read their answers to the comments of other reviewers. In particular, the answers to the comments on the novelty of their work are convincing. Therefore, I can recommend the paper for publication in Nature Communications.